# Discriminating neural ensemble patterns through dendritic computations in randomly connected feedforward networks

**Bhanu Priya Somashekar, Upinder Singh Bhalla***

National Centre for Biological Sciences, Tata Institute of Fundamental Research, Bangalore, India

## eLife Assessment

This study presents **valuable** quantitative insights into the prevalence of functionally clustered synaptic inputs on neuronal dendrites. The simple analytical calculations and computer simulations provide **solid** support for the main arguments. The findings can lead to a more detailed understanding of how dendrites contribute to the computation of neuronal networks.

*For correspondence:
bhalla@ncbs.res.in

**Abstract** Co-active or temporally ordered neural ensembles are a signature of salient sensory, motor, and cognitive events. Local convergence of such patterned activity as synaptic clusters on dendrites could help single neurons harness the potential of dendritic nonlinearities to decode neural activity patterns. We combined theory and simulations to assess the likelihood of whether projections from neural ensembles could converge onto synaptic clusters even in networks with random connectivity. Using rat hippocampal and cortical network statistics, we show that clustered convergence of axons from three to four different co-active ensembles is likely even in randomly connected networks, leading to representation of arbitrary input combinations in at least 10 target neurons in a 100,000 population. In the presence of larger ensembles, spatiotemporally ordered convergence of three to five axons from temporally ordered ensembles is also likely. These active clusters result in higher neuronal activation in the presence of strong dendritic nonlinearities and low background activity. We mathematically and computationally demonstrate a tight interplay between network connectivity, spatiotemporal scales of subcellular electrical and chemical mechanisms, dendritic nonlinearities, and uncorrelated background activity. We suggest that dendritic clustered and sequence computation is pervasive, but its expression as somatic selectivity requires confluence of physiology, background activity, and connectomics.

## Introduction

Neural activity patterns in the form of sequential activity or co-active ensembles reflect the structure of environmental and internal events across time, stimuli, and representations. For example, sequential auditory activity may signify predators, mates, or distress calls. How can combinations of active ensembles at an input layer be discriminated by downstream neurons?

Dendrites offer a repertoire of nonlinearities that could aid neurons in achieving such computations (*Ranganathan et al., 2018*; *Lee et al., 2016b*; *Palmer et al., 2014*). Several of these mechanisms can be triggered by the local convergence of inputs onto dendritic segments. For example, clustered inputs may trigger dendritic spikes (*Gasparini and Magee, 2006*; *Gasparini et al., 2004*). They

could trigger $Ca^{2+}$ release from internal stores (*Lee et al., 2016b*) and other biochemical signaling cascades. Dendritic $Ca^{2+}$spikes have been shown to enhance the mixing of sensory (whisker touch) and motor (whisker angle) features in the pyramidal neurons of layer 5 vibrissal cortex (*Ranganathan et al., 2018*). If local convergence of interesting inputs is a prerequisite for engaging such dendritic nonlinearities, a key question that remains is whether networks require specialized wiring rules to discriminate ensemble activity patterns, or whether such localized connectivity is likely to arise even with purely random connectivity.

There is considerable evidence for clustered synaptic activity in several brain regions, including the visual cortex (*Gökçe et al., 2016*; *Wilson et al., 2016*), hippocampus (*Kleindienst et al., 2011*; *Adoff et al., 2021*), and motor cortex (*Kerlin et al., 2019*). A recent study done in macaques showed that synaptic inputs on dendrites are spatially clustered by stimulus features such as orientation and color (*Ju et al., 2020*). In the hippocampus, it has been shown that place cells receive more temporally co-active, clustered excitatory inputs within the place field of the soma than outside (*Adoff et al., 2021*). Notably, the length scale of these clustered inputs is ~10 μm, similar to the scale of operation of local calcium-induced calcium release (CICR) on the dendrite (*Lee et al., 2016b*; *O'Hare et al., 2022*).

Many salient events are ordered in time, giving rise to convergent inputs which are not just co-active, but ordered sequentially in space and time on the dendrite. This form of dendritic computation was predicted by *Rall, 1964*. Branco and colleagues have shown that pyramidal neurons can discriminate spatiotemporal input patterns within 100 μm dendritic zones (*Branco et al., 2010*). *Bhalla, 2017*, employed reaction-diffusion models of signaling molecules to show the emergence of sequence selectivity in small dendritic zones. It is therefore valuable to analyze the statistical constraints on connectivity and input activity that can lead to synaptic clustering and dendritic sequence discrimination.

Seminal analytical work from *Poirazi and Mel, 2001*, showed that dendrites can increase the memory capacity of neurons by ~23-fold by combining subsets of synaptic inputs nonlinearly. There has also been an attempt to estimate the likelihood of organization of clustered inputs using combinatorial approaches (*Scheuss, 2018*). In the current study, we consider multiple different dimensions, including connection probability, ensemble activity, spatiotemporal scales, dendritic nonlinearities, and the role of uncorrelated noise to explore the conditions under which grouped and sequential dendritic computation may emerge from random connectivity. We show that convergent connectivity of features to three to five synapses are statistically likely to occur on short dendritic segments in feedforward networks of ~100,000 neurons with random connectivity. Further, these clusters may perform combinatorial input selection and sequence recognition at the dendritic level. These result in somatic selectivity under conditions of low network noise, high co-activity in upstream neurons, and strong dendritic nonlinearities.

## Results

We examined the mapping of network functional groupings, such as ensembles of co-active neurons onto small, <100 μm zones of synapses on neuronal dendrites. We derived approximate equations for convergence probabilities for a randomly connected feedforward network, examined the effect of noise, and then supported these estimates using connectivity-based simulations. Next, we tested the effect of noise on dendritic mechanisms for sequence selectivity using electrical and chemical models. Finally, we combined these constraints to examine the likely sequence lengths that may emerge in random networks.

We framed our analysis as a two-layer feedforward network in which the input layer is organized into several ensembles within which cells are co-active in the course of representing a stimulus or other neural processing (colored dots in the presynaptic population in *Figure 1A and C*). In addition to neurons representing ensemble activity, the input layer also contained neurons that represented background activity/noise, whose activity was uncorrelated with the stimulus (gray dots in the presynaptic population in *Figure 1C*). The 'output' or 'postsynaptic' layer was where we looked for convergence of inputs. We examined the probability of projections from specific ensembles onto small local dendritic zones on a target neuron (*Figure 1A, B, and C*). Within this framework, we considered two cases: first, that projections are grouped within the target zone without any spatiotemporal order (*grouped* input). Second, that these projections are spatially ordered in the same sequence as the activity of successive ensembles (*sequential* input) (*Figure 1B*).

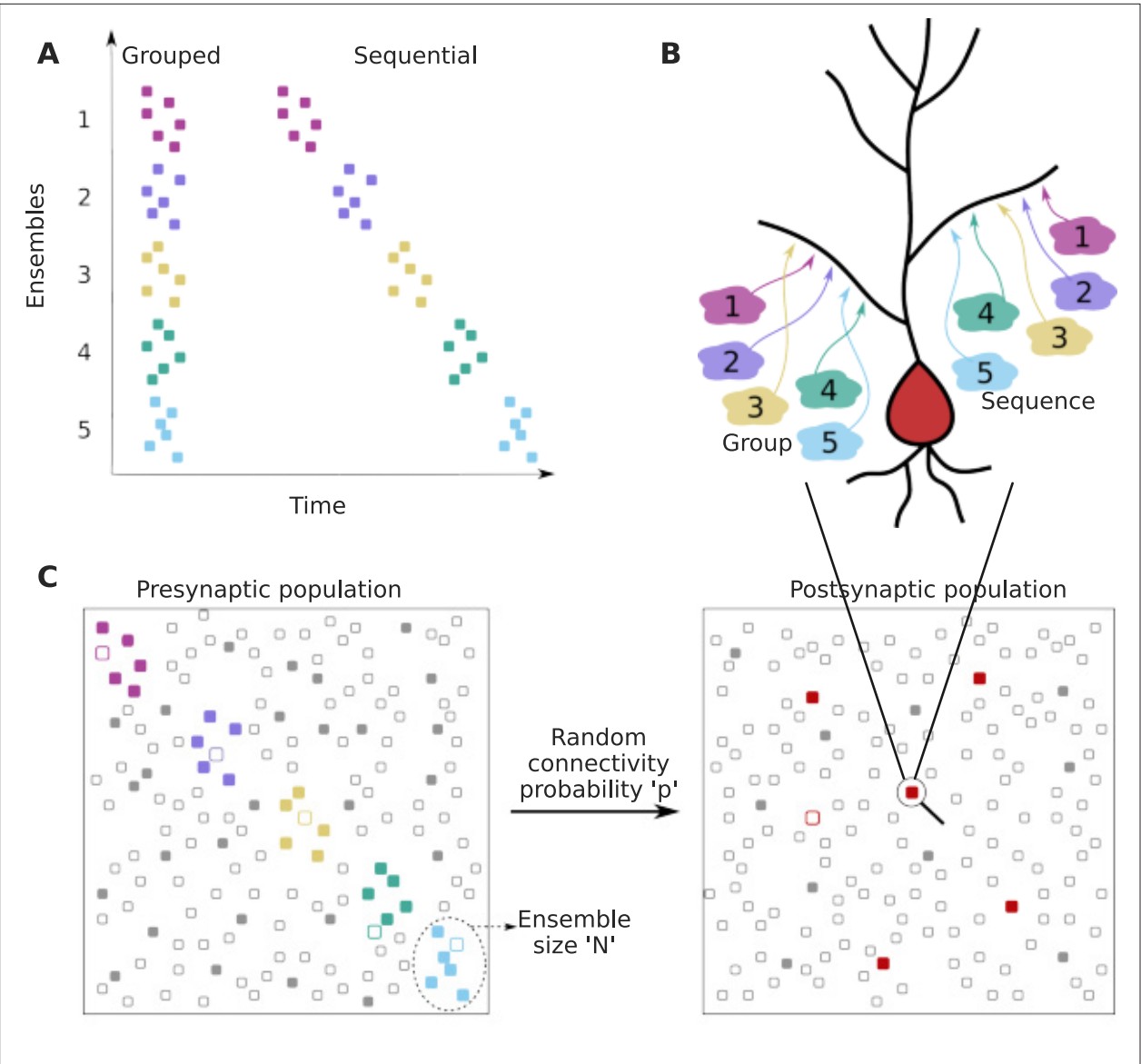

**Figure 1.** Dendritic convergence of inputs from neural ensembles as groups and sequences. (**A**) Schematic of ensemble activity for grouped (left) and sequential (right) activity patterns. In this illustration there are five distinct ensembles represented using different colors, each with six neurons. In grouped activity all the ensembles are active in the same time window, whereas in sequential activity the ensembles are active successively. (**B**) Schematic of projection patterns of grouped (left) and sequential (right) connectivity patterns. There is no spatial organization of the grouped inputs, other than their convergence to a small target zone on the dendrite. Sequential inputs also converge to a small target zone, but connect to it in a manner such that the temporal ordering gets mapped spatially. (**C**) Schematic of pre- and postsynaptic populations. The postsynaptic population receives connections from the presynaptic population through random connectivity. Colored dots in the presynaptic population represent ensembles, whereas gray dots represent neurons that are not part of ensembles, but may participate in background activity. Unfilled/filled dots in the presynaptic population represent neurons that are inactive/active respectively, within the temporal window of interest. The red dots in the postsynaptic population represent neurons that receive grouped/sequential connections from ensembles on their dendrites in addition to regular ensemble and background inputs, while gray dots represent those that do not receive such organized connections. Filled red dots represent neurons that receive all synaptic inputs constituting a group/sequence. Unfilled red dots are anatomically connected to a group/sequence, but some of their presynaptic neurons may be inactive in the considered time window. Filled gray dots are neurons that receive groups/sequences from background inputs either purely or in combination with some ensemble inputs, whereas unfilled gray dots represent neurons that do not receive grouped/sequential inputs. The population activity shown using filled neurons represents neurons that are active within the duration of a single occurrence of grouped/sequential activity.

**Table 1.** Example network configurations and parameters.

| Name | p, Probability of connectivity | $T_{pre}$, #neurons in the local network | R, Background input rate (Hz) | D, Time window for each input (s), also corresponds to the timescale of ensemble activity | Z, Zone length (for groups) (µm) | S, Spacing between inputs (for sequences) (µm) | Δ, Delta, Available window for convergence of input (for sequences) (µm) |
|---|---|---|---|---|---|---|---|
| Hippo-chem | 0.05 | 400,000 | 0.01 | 2 | 10 | 2 | 1.5 |
| Hippo-CICR | 0.05 | 400,000 | 0.01 | 0.2 | 10 | 2 | 1.5 |
| Hippo-elec | 0.05 | 400,000 | 0.1 | 0.004 | 50 | 10 | 5 |
| Cortex-chem | 0.2 | 100,000 | 0.1 | 2 | 10 | 2 | 1.5 |
| Cortex-CICR | 0.2 | 100,000 | 0.1 | 0.2 | 10 | 2 | 1.5 |
| Cortex-elec | 0.2 | 100,000 | 1 | 0.004 | 50 | 10 | 5 |

The dendrite may engage different postsynaptic mechanisms at different spatiotemporal scales to mediate computation and plasticity among these inputs. We looked at three such classes of mechanisms: *electrical*, *chemical*, and *calcium-induced calcium release (CICR)*. Electrical mechanisms operate at a spatial scale of 50–100 µm and work on a faster timescale of <50 ms (***Golding et al., 2002***; ***Brandalise et al., 2016***; ***Branco et al., 2010***; ***Gasparini and Magee, 2006***). Chemical mechanisms work over behavioral timescales of a few seconds and ~10 µm range since they are limited by the diffusion of proteins within the dendrite (***Bhalla, 2017***). CICR is a mechanism by which $Ca^{2+}$ influx from external sources triggers the release of $Ca^{2+}$ from intracellular stores thus amplifying cytosolic $Ca^{2+}$ concentration. This process typically operates on a length scale of ~10 µm and a timescale of ~100–1000 ms (***Zhou and Ross, 2002***; ***Nakamura et al., 2002***). It is mediated by the inositol trisphosphate and ryanodine receptors present on the endoplasmic reticulum. The length scales of CICR and chemical computation for grouped inputs are consistent with what are typically referred to as 'synaptic clusters' in the field.

For our study we chose a total of six representative circuit configurations, considering hippocampal and cortical connectivity and firing statistics for each of the three above-mentioned dendritic integration mechanisms (***Table 1***).

We used these broad categories as representatives to explore possible regimes for grouped and sequence computation in feedforward networks with random connectivity (***Figure 1***).

## Grouped convergence of inputs

Clustered input is capable of eliciting stronger responses than the same inputs arriving in a dispersed manner (***Polsky et al., 2004***; ***Schiller et al., 2000***; ***Gasparini and Magee, 2006***). In principle, several electrical and signaling mechanisms with high order reactions or high cooperativity support such selectivity (***Schiller et al., 2000***; ***Kumar et al., 2018***; ***Bhalla, 2003***). We illustrated this using a simple model of Calmodulin activation by $Ca^{2+}$ influx (***Figure 2A***) from grouped and spaced synapses, respectively. Grouped inputs arriving at a spacing of 2 µm resulted in a higher concentration of activated Calmodulin (CaM-Ca4) when compared to inputs that arrived in a dispersed manner, at a spacing of 10 µm (***Figure 2B***). Since grouped inputs can indeed lead to larger responses at the postsynaptic dendritic branch, we investigated the likelihood of inputs converging in a grouped manner in the presence of random connectivity.

## Sparse representation of fully mixed groups of three to four inputs is likely in randomly connected networks

We formulated an analytical expression to estimate the bounds for the probability of convergence for grouped inputs in feedforward networks with random connectivity. Our approach is to ask how likely it is that a given set of inputs lands on a short segment of dendrite, and then scale it up to all segments on the entire dendritic length of the cell. Consider a neuron of total dendritic length L. L is in the range of 10 mm for CA1 pyramidal neurons in rats (***Vitale et al., 2023***; ***Bannister and Larkman,***

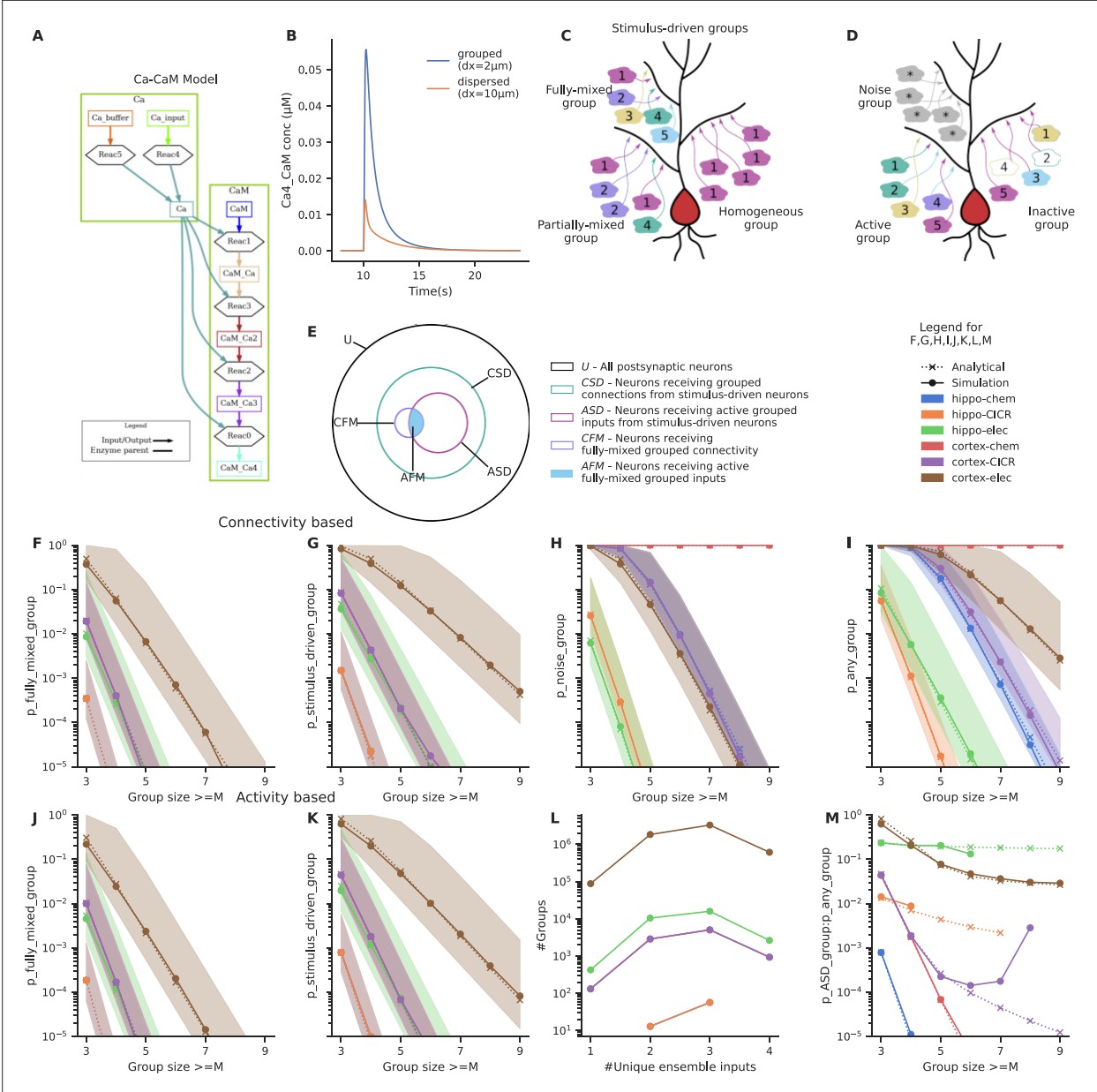

**Figure 2.** Grouped convergence of noisy and stimulus-driven synaptic inputs is likely in all network configurations. (**A**) A simple, biologically inspired model of Ca-Calmodulin reaction pathway. (**B**) The Ca-Calmodulin model shows selectivity for grouped inputs. Inputs arriving in a grouped manner (2 μm spacing) lead to a higher concentration of Ca4_CaM as opposed to dispersed inputs (10 μm spacing). (**C**) Different kinds of stimulus-driven groups. 'Stimulus-driven groups' receive M or more connections from neurons belonging to any of the ensembles. They could be of three kinds: (1) 'fully mixed groups' that receive at least one connection from each ensemble; (2) 'partially mixed groups' that contain multiple inputs from the same ensemble while missing inputs from others; 'homogeneous groups' that receive inputs from a single ensemble. (**D**) Active and inactive groups, and noise groups. In an active group, all inputs constituting a group are received, whereas an inactive group may be missing one or more inputs in the duration of occurrence of a group, in spite of being connected with the ensembles. 'Noise groups' are formed of M or more background inputs; a group composed of M or more inputs of any kind, either from the ensembles or from noise or a combination of both is referred to as 'any group'. (**E**) Neurons classified as per the types of groups they receive. Note that the schematic is a qualitative representation. The sizes of the circles do not correspond to the cardinality of the sets. (**F**) Probability of occurrence of connectivity-based fully mixed group due to local convergence of ensemble inputs. Group size is the number of different ensembles. It also corresponds to the number of different ensembles that send axons to a group of zone length 'Z'. The calcium-induced calcium release ('CICR') configuration overlaps with the 'chem' configuration of the corresponding network in **F, G, J** and **K** as they share the same zone length. (**G**) Probability of occurrence of connectivity-based stimulus-driven group, which receives connections from any of the ensembles. Group size in G refers to the number of connections arriving from any of the ensemble neurons within zone length 'Z'. (**H**) Probability of occurrence of noise group due to local noisy activation of synapses. Here, the group size refers to the number of synapses activated within the zone of length 'Z'. (**I**) Probability

*Figure 2 continued on next page*

*Figure 2 continued*

of occurrence of a group due to the convergence of either stimulus-driven inputs or noisy inputs or a combination of both. Hippo-chem overlaps with cortex-CICR as they have same value for R*D. (**J, K**) Probability of occurrence of active fully mixed and active stimulus-driven groups respectively, wherein all inputs constituting a group are active. Shaded regions in **F, G, H, I, J, K** represent lower and upper bounds on the analytical equations for probability based on non-overlapping and overlapping cases, i.e., $\kappa = \frac{L}{Z}$ and $\kappa = \frac{L}{\sigma}$ in the equations. (**L**) Frequency distribution of groups based on the number of unique ensembles they receive connections from, for stimulus-driven groups that receive four or more ensemble inputs. Here, the total number of active ensembles in the presynaptic population is four. (**M**) Ratio of the probability of occurrence of an active stimulus-driven group to the probability of occurrence of any group on a neuron. This gives an indirect measure of signal to noise in the population. The mismatch seen between the analytical and simulated traces in the case of cortex-CICR is due to low sampling at higher group sizes.

The online version of this article includes the following figure supplement(s) for figure 2:

**Figure supplement 1.** Factors affecting the likelihood of grouped convergence for different kinds of groups.

**Figure supplement 2.** Probability of occurrence of small groups that receive inputs from two ensembles.

*1995*), which we collapse into a single long notional dendrite. Our analysis does not consider electrotonic effects, hence all synapses are equivalent. Assume activity of M ensembles, each containing N neurons. We refer to groups of synapses within the target zone receiving at least one connection from each ensemble as *connectivity-based fully mixed groups (cFMG)*, because they hold the potential to associate information from all these ensembles (*Figure 2C*). We seek the probability $P_{cFMG}$, that there is at least one synaptic projection from each of the M ensembles to a target zone of length Z (*a group*) occurring anywhere on a single target neuron of total dendritic arbor length L. The parameter p is the probability of any given neuron from the presynaptic population to connect onto a given postsynaptic cell. Values of p for the hippocampal CA3-CA1 projections have been estimated at 5% (*Bolshakov and Siegelbaum, 1995*; *Sayer et al., 1990*), and for cortical circuits in the range of 5–50% (*Brown and Hestrin, 2009*; *Holmgren et al., 2003*; *Ko et al., 2011*; *Table 1*, *Table 2*).

Expectation number of connections from any neuron in a particular ensemble, anywhere on target cell = $pN$.

Expectation number of inputs from any neuron in a particular ensemble, on a zone of length Z on the target cell = $\frac{pNZ}{L} = \nu_{cFMG}$.

Probability of a zone of length Z receiving one or more inputs from a particular ensemble $\approx 1 - e^{-\nu_{cFMG}}$.

**Table 2.** Parameters used in the analytical derivations.

| Parameter | Represents |
|---|---|
| p | Connection probability, i.e., probability of a neuron in the postsynaptic population being connected with a neuron in the presynaptic population |
| $p_e$ | Ensemble participation probability, i.e., probability that a neuron part of an ensemble is active in a single occurrence of the stimulus. |
| N | Number of neurons in an ensemble |
| L | Total dendritic arbor length of a postsynaptic neuron |
| Z | Zone length for dendritic group computation |
| M | Number of ensembles; also corresponds to the number of inputs relevant for grouped computation (in certain scenarios) or sequence computation |
| $T_{pre}$ | Number of neurons in the presynaptic population |
| $T_{post}$ | Number of neurons in the postsynaptic population |
| σ | Inter-synapse interval; also equal to $T_{pre}*p$ |
| S | Minimum spacing between two inputs that are are part of a sequence |
| Δ | Available window where the next input can arrive to combine sequentially with the previous input |
| D | Duration of correlated activity among neurons constituting an ensemble |
| R | Rate of background activity in the presynaptic population |
| $p_{bg}$ | Probability of a synapse being hit by background activity in duration D; equal to $1-e^{-RD}$ |

Probability of a zone of length Z receiving one or more inputs from each of the M ensembles = $P_{cFMG\_Z} \approx \left(1 - e^{-\nu_{cFMG}}\right)^{M}$.

The probability of occurrence of at least one zone of length Z anywhere along on the dendritic arbor, such that it receives at least one input from each ensemble is given by:

$$P_{cFMG} \approx 1 - \left(1 - P_{cFMG\_Z}\right)^{\kappa_{cFMG}} \qquad (1)$$

where $\kappa_{cFMG}$ lies between $\frac{L}{Z}$ and $\frac{L}{\sigma}$ depending on degree of overlap across zones in different network configurations. Here, σ denotes the inter-synapse interval which is 0.5 μm in all our network configurations. In *Figure 2F*, we compare the analytical form obtained by fitting *Equation 1* with κ as the free parameter to the probabilities calculated from our connectivity-based simulations (check Methods). The values of $\kappa_{cFMG}$ vary for different network configurations (*Supplementary file 1c*).

Thus, the probability of occurrence of groups that receive connections from each of the M ensembles ($P_{cFMG}$) is a function of the connection probability (p) between the two layers, the number of neurons in an ensemble (N), the relative zone length with respect to the total dendritic arbor (Z/L), and the number of ensembles (M).

Ensemble neurons may not be 100% reliable in representing a stimulus. If we assume that each ensemble neuron is active with a probability $p_e$ during the presence of the corresponding stimulus, a connected fully mixed group may not always receive all inputs that constitute the group in a single occurrence of the stimulus (*Figure 2D*). Hence, the probability of a neuron receiving an *active fully mixed group* where there is at least one active input arriving from each of the M ensembles is given by

$$P_{aFMG} \approx 1 - \left(1 - P_{aFMG\_Z}\right)^{\kappa_{aFMG}} \qquad (2)$$

where

$$P_{aFMG\_z} \approx (1 - e^{-\nu_{aFMG}})^{M} \; and \; \nu_{aFMG} = \frac{pp_e NZ}{L}$$

where $\kappa_{aFMG}$ again lies between $\frac{L}{Z}$ and $\frac{L}{\sigma}$ depending on the degree of overlap between zones.

We validated these analytic expressions using simulations of network connectivity as described in the Methods (*Figure 2*). The simulations checked for the occurrence of such groups on each neuron in the postsynaptic population simulated through random connectivity. The simulation estimates matched the analytic expressions.

To interpret these relationships (*Figure 2*) we stipulate that pattern discrimination via dendritic grouped computation requires that at least one neuron in the downstream population receives convergent connections from all the M ensembles. Hence, if $P_{cFMG} < 1/T_{post}$, where $T_{post}$ = the total number of neurons in the postsynaptic population, such projections are unlikely to happen anywhere in the network. Conversely, if $P_{cFMG} \sim 1$, then all the neurons in the postsynaptic population receive similar grouped inputs, leading to very high redundancy in representation. A sparse population code would be ideal for this scenario with a few neurons receiving such convergent grouped inputs from each pattern of ensemble activity. With ensembles of 100 neurons in the presynaptic population, we find that in a downstream population of 100,000 neurons, fully mixed groups of three to four inputs are likely to occur in four out of the six network configurations we tested (hippo-elec, cortex-chem, cortex-CICR, cortex-elec) (*Figure 2F*). Since $P_{aFMG}$ is exponentially related to M, the probability of groups with greater number of inputs drops steeply as the number of ensembles increases. Large fully mixed groups of up to seven inputs are likely only in the cortex-electrical configuration which has higher connection probability and longer zone length. Thus, grouped convergence of three to four inputs from a specific neural activity pattern seems likely as well as potentially non-redundant, in most network configurations.

## End-effects limit convergence zones for highly branched neurons

Neurons exhibit considerable diversity with respect to their morphologies. How synapses extending across dendritic branch points interact in the context of a synaptic cluster/group is a topic that needs detailed examination via experimental and modeling approaches. However, for the sake of analysis, we present calculations under the assumption that selectivity for grouped inputs might be degraded across branch points.

Zones beginning close to a branch point might get interrupted. Consider a neuron with B branches. The length of the typical branch would be L/B. As a conservative estimate if we exclude a region of length Z for every branch, the expected number of zones that begin too close to a branch point is

$$E_{end} \approx \frac{ZB}{L}$$ (3)

For typical pyramidal neurons B~50, so $E_{end}$~0.05 for values of Z of ~10 µm. Thus, pyramidal neurons will not be much affected by branching effects. Profusely branching neurons like Purkinje cells have B~900 for a total L of ~7800 µm (*McConnell and Berry, 1978*), hence $E_{end}$~1 for values of Z of ~10 µm. Thus, almost all groups in Purkinje neurons would run into a branch point or terminal. For the case of electrical groups, this estimate would be scaled by a factor of 5 if we consider a zone length of 50 µm. However, it is important to note that these are very conservative estimates, as for clusters of four to five inputs, the number of synapses available within a zone are far greater (~100 synapses within 50 µm).

## Random connectivity supports stimulus-driven groups that mix information from different ensembles to varying extents

Since ensembles typically consist of more than one neuron which show correlated activity, this could lead to neurons receiving different combinations of ensemble inputs. For example, neurons could receive *fully mixed groups* that associate inputs from five different ensembles {A, B, C, D, E}, or *partially mixed groups* such as {A, B, A, A, D} where inputs from one or more active ensembles are missing, or *homogeneous groups* such as {B, B, B, B, B} which are still correlated and contain the same number of inputs, but all inputs come from a single ensemble. We refer to the superset consisting of all these different kinds of groups as *stimulus-driven groups* (*Figure 2C*).

With presynaptic ensembles of 100 neurons, stimulus-driven groups are more frequent than fully mixed groups (compare *Figure 2F* with 2G and 2J with 2K; derivations in Appendix 1). This gap increases further as the group size increases owing to the increasing number of permutations possible with repeating inputs (compare slopes of *Figure 2F and G*). We found that even with ensembles of 100 neurons, there is enough redundancy in input representation such that the combinatorics yield a greater proportion of partially mixed groups, relative to fully mixed and homogeneous groups (*Figure 2L*).

## Background activity can also contribute to grouped activity, resulting in false positives

Networks are noisy, in the sense of considerable background activity. False positives arise when noisy inputs combine with other noisy inputs or with one or more inputs from the ensembles to trigger the same dendritic nonlinearities as true positives. We assumed the background activity to be Poisson in nature, with synapses receiving inputs at a rate R. We chose R in the range of random background activity, ~0.1 Hz for CA3 pyramidal neurons and ~1 Hz for the prefrontal cortex (*Table 1*). For chemical inputs we require strong, Ca-influx triggering bursts, which are likely to be rare. Here, we assume that R is about 10% of the network background activity rate, or ~0.01 Hz for CA3 and 0.1 Hz for cortex (*Buzsáki and Mizuseki, 2014*; *Mizuseki and Buzsáki, 2013*).

Assuming Poisson statistics, for a network firing at a rate of R Hz, the probability of a synapse being hit by background activity in duration D is $1-e^{-RD}$. With this we estimated the probability of occurrence of *noise groups* which receive grouped inputs from background activity alone (*Figure 2H*), and *any groups* which receive either background inputs or ensemble inputs or combination of both in a grouped manner (*Figure 2I*) (derivations in Appendix 1). Noise groups are more probable in the cortical configurations than hippocampal configurations owing to the higher rate of background activity. The larger time window for integration in the cortex-chem case worsens the situation as the probability for noise groups is ~1 for most group sizes tested. Further, smaller noise groups (with group size of 3–5) are much more likely than larger ones.

The influence of various parameters on the probabilities of different kinds of groups is examined in *Figure 2—figure supplement 1* using cortex-CICR as an example, varying one parameter at a time. Dense connectivity, longer zone length, and larger ensembles all increase the probability of occurrence of fully mixed and stimulus-driven groups.

Based on these calculations, it would be near impossible to identify neurons receiving stimulus-driven groups from other neurons in the population under high noise conditions (compare cortex-chem in *Figure 2K* with 2H and 2I). A comparison of the probability of active stimulus-driven groups to the probability of any groups shows that the hippo-elec configuration is the best suited for grouped computation as it has the best 'signal-to-noise' ratio. Hence, we chose the hippo-elec as a case study below, to understand how dendritic nonlinearities influence neuronal output in our postsynaptic neurons.

## Strong dendritic nonlinearities are necessary to distinguish neurons receiving stimulus-driven groups from other neurons

What significance would grouped activity on a small dendritic zone have on neuronal output, given multiple other inputs synapsing onto the dendritic arbor? The strength of dendritic nonlinearity could influence the relative contribution of grouped inputs to the output of a neuron. We evaluated how nonlinearities could distinguish neurons that receive stimulus-driven groups from other neurons in the population based on their somatic activity profiles.

We applied a nonlinear function over the number of inputs converging within a zone of length Z, using a sliding window. This provides a measure of the local activation produced due to the nonlinear combining of grouped inputs. We integrated these local activations across the entire length of the dendrite, as a proxy for somatic activation. We chose a function inspired from the cumulative of the Weibull distribution (equation shown in *Figure 3A*) to capture the nonlinear integration of grouped inputs (*Figure 3A*). This function has several interesting properties: it is smooth, nonlinear, saturates at high input numbers, and most importantly the power controls the strength of contribution that small groups have relative to the large groups. We varied the power term $\rho$ to control the strength of the nonlinearity for the case of the hippo-elec configuration to distinguish neurons receiving groups containing four or more inputs.

In the presence of weak nonlinearity ($\rho$ =2) neurons receiving grouped connectivity from stimulus-driven neurons (CSD neurons) show activation values that are similar to neurons that do not receive such groups (CSD$^C$ neurons) (*Figure 3B and D*). However, when the strength of the nonlinearity is increased, the CSD neurons can be easily distinguished from CSD$^C$ neurons as strong activations arise only in the presence of large groups (group size $\geq$ 4) (*Figure 3C and F*). Under conditions of weak nonlinearity, the kinds of activation profiles seen in certain fractions of trials are also similar for CSD and CSD$^C$ neurons (*Figure 3H and I*). In this case, even with a more nuanced measurement than trial-averaged activation one will not be able to distinguish the two types of neurons. In contrast, in the presence of strong nonlinearity one finds a clear regime where CSD neurons show higher activation values (>0.13) in a larger fraction of trials (>0.15) (red dashed region) when compared to CSD$^C$ neurons (*Figure 3J and K*).

Even in the presence of strong nonlinearities, neurons receiving fully mixed grouped connectivity (CFM neurons) cannot be distinguished from their complement set (CFM$^C$ neurons) (*Figure 3E and G*). This is because of the large number of neurons in the CFM$^C$ group that receive either partially mixed groups or homogeneous groups, and therefore show similar activation profiles.

In summary, we find that strong nonlinearities are necessary to distinguish the activity of neurons that receive relevant grouped connectivity from other neurons in the network. However, even in the presence of strong nonlinearities, it is not possible to distinguish neurons that receive fully mixed groups from partially mixed/homogeneous groups, on the basis of responses to a single pattern of ensemble activations.

## Sequential convergence of inputs
### Sequences of three to five inputs are likely, but require the presence of larger ensembles

We next performed a similar calculation with the further requirement that the ensembles are active in a certain order in time, and that the projections arrive on the target neuron in the same spatial order as their activity. This form of computation has been observed for electrical discrimination of sequences (*Branco et al., 2010*) and has been proposed for discrimination based on chemical signaling (*Bhalla, 2017*). Here, we estimate the likelihood of the first ensemble input arriving anywhere on the dendrite, and ask how likely it is that succeeding inputs of the sequence would arrive within a set spacing. We

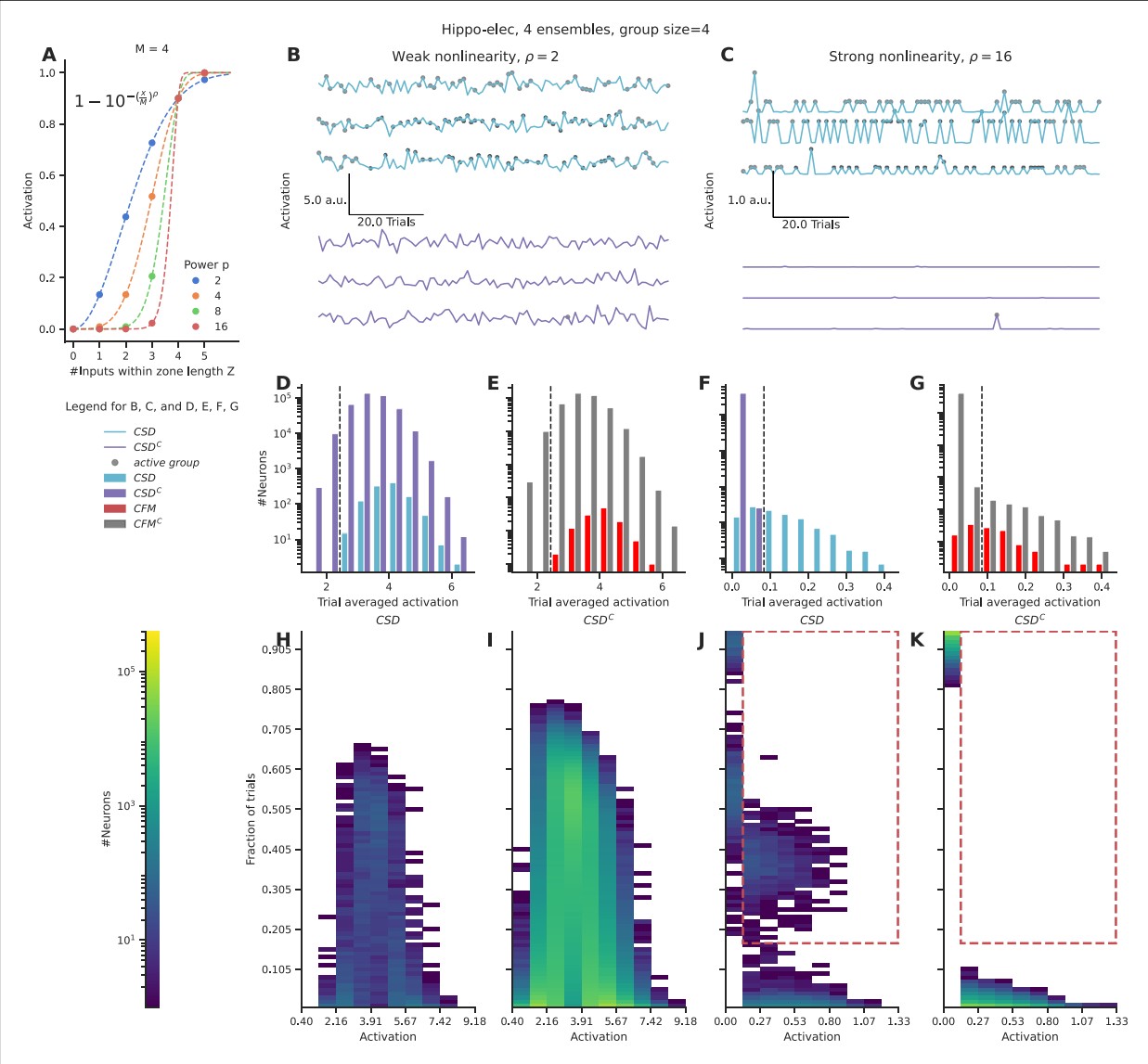

**Figure 3.** Strong nonlinearities help distinguish neurons receiving stimulus-driven groups from other neurons. (**A**) Dendritic activation, measured as the nonlinear integration of inputs arriving within the dendritic zone Z using the formulation mentioned in the inset. (**B, C**) Representative total neuronal activation across 95 trials of three neurons that receive grouped connections from stimulus-driven neurons (CSD) (cyan) and three neurons that do not receive grouped connections from stimulus-driven neurons (CSD$^C$) (violet) under conditions of weak nonlinearity (**B**) vs strong nonlinearity (**C**). Gray dots mark the occurrence of an active group of any kind. (**D, E**) Trial-averaged neuronal activation of CSD vs CSD$^C$ neurons, and neurons receiving connectivity-based fully mixed groups vs neurons not receiving connectivity-based fully mixed groups, respectively, in the presence of weak nonlinearity. (**F, G**) Similar to D, E, but in the presence of strong nonlinearity. The dashed black line in D, E, F, G represents an example threshold that would help discriminate CSD neurons from CSD$^C$ neurons based on trial-averaged activation values. Strong nonlinearity helps discriminate CSD neurons from CSD$^C$ neurons. However, CFM neurons cannot be distinguished from CFM$^C$ neurons even in the presence of a strong nonlinearity. (**H, I, J, K**) Number of neurons showing activation values within the specified range in a certain number of trials. (**H, J**) represent CSD neurons under conditions of weak and strong nonlinearities respectively. (**I, K**) represent CSD$^C$ under conditions of weak and strong nonlinearities respectively. CSD$^C$ neurons show similar activation profiles in similar fractions of trials when compared to CSD neurons. This combined with the large population size of the CSD$^C$ neurons relative to CSD neurons, makes it almost impossible to discriminate between the two groups under conditions of weak nonlinearity. In the presence of strong nonlinearities, CSD neurons show higher activation profiles (0.13–1.33 a.u.) in a higher percentage of trials (>15%) (the region marked with the red dashed line). Hence, they can be discriminated from CSD$^C$ neurons in the presence of a strong nonlinearity.

stipulate that each successive input lies within a spacing of S to S+Δ μm with respect to the previous input. We assume that each ensemble is active over a duration D. Mean spine spacing is 0.3–1 μm in rat and mouse pyramidal neurons both in cortex and hippocampus (***Bannister and Larkman, 1995***; ***Konur et al., 2003***). The typical spacing S between inputs is of the order of 2 μm for chemical/CICR

sequences and ~10 µm for electrical sequences. We assume Δ to be ~1.5 µm for chemical/CICR and ~5 µm for electrical sequences. Hence, there can be intervening synapses between the stimulated ones.

First, we estimate the probability of *connectivity-based perfectly ordered stimulus sequences* (cPOSS), which receive connections in an ordered manner from each of the ensembles (*Figure 4A and B*).

Expected number of connections from neurons in the first ensemble, anywhere on a given postsynaptic neuron = $pN$.
Expected number of connections from neurons in the second ensemble within a spacing of S to S+Δ from a connection coming from the previous ensemble = $\nu_{cPOSS} = \frac{pN\Delta}{L}$.
Expectation number of sequences containing a connection from the first ensemble followed by an connection from the second ensemble, within a spacing of S to S+Δ with respect to the first ensemble input = $pN\nu_{cPOSS}$.
Expectation number of M-length perfect sequences, that receive sequential connectivity from stimulus-driven neurons arriving from M different ensembles following the above spacing rule, on a neuron = $E_{cPOSS} = pN\left(\nu_{cPOSS}\right)^{M-1}$.

Using Poisson approximation, the probability of one or more cPOSS which receive axons from M different ensembles, occurring anywhere along the dendritic length of a neuron, is

$$P_{cPOSS} \approx 1 - e^{-E_{cPOSS}} \tag{4}$$

Thus, the probability of occurrence of sequences that receive sequential connections ($P_{cPOSS}$) from each of the M ensembles is a function of the connection probability ($p$) between the two layers, the number of neurons in an ensemble ($N$), the relative window size with respect to the total dendritic arbor ($\Delta/L$), and the number of ensembles ($M$).

Here, we assume a specific ordering of at least one projection from each of the M ensembles. The original calculations (*Bhalla, 2017*) for chemical sequence selectivity were symmetric, thus the ordering could be distal-to-proximal, or vice versa. This would introduce a factor of 2 if there were no other source of symmetry breaking in the system:

$$2pN\left(\nu_{cPOSS}\right)^{M-1} = E_{cPOSS}$$

For the purposes of further analysis, we assume that the neuronal system is not symmetric and hence the ordering remains unidirectional as in *Equation 4*.

If one considers only *activity-based perfectly ordered stimulus sequences* (aPOSS) (*Figure 4B*) where all inputs constituting a sequence are received in a single instance of sequential stimulation of the ensembles, we get

$$P_{aPOSS} \approx 1 - e^{-E_{aPOSS}} \tag{5}$$

where $E_{aPOSS} = p_e pN\left(\frac{p_e pN\Delta}{L}\right)^{M-1}$, assuming that neurons in the ensemble participate with a probability of $p_e$.

When the additional constraint of order was imposed, much larger ensembles (~1000 neurons), as opposed to 100 neurons in the case of groups, were required to obtain sequential connectivity (*Figure 4D*) and sequential activation (*Figure 4G*) of three to five inputs in a population of ~100,000 neurons. Hence, sequence discrimination required greater redundancy in input representation in the form of larger ensembles.

## Background activity can fill gaps in partial sequences to yield false positives

We next estimated the probability of occurrence of different kinds of sequences: *noise sequences* which receive background inputs alone (*Figure 4A*), *any sequences* which receive either background inputs or ensemble inputs or a combination of both, and *gap-fill sequences* which contain one more ensemble inputs that make up a partial sequence, with the missing inputs being filled in by background activity arriving in the right location at the right time (*Figure 4A*) (derivations in Appendix 1).

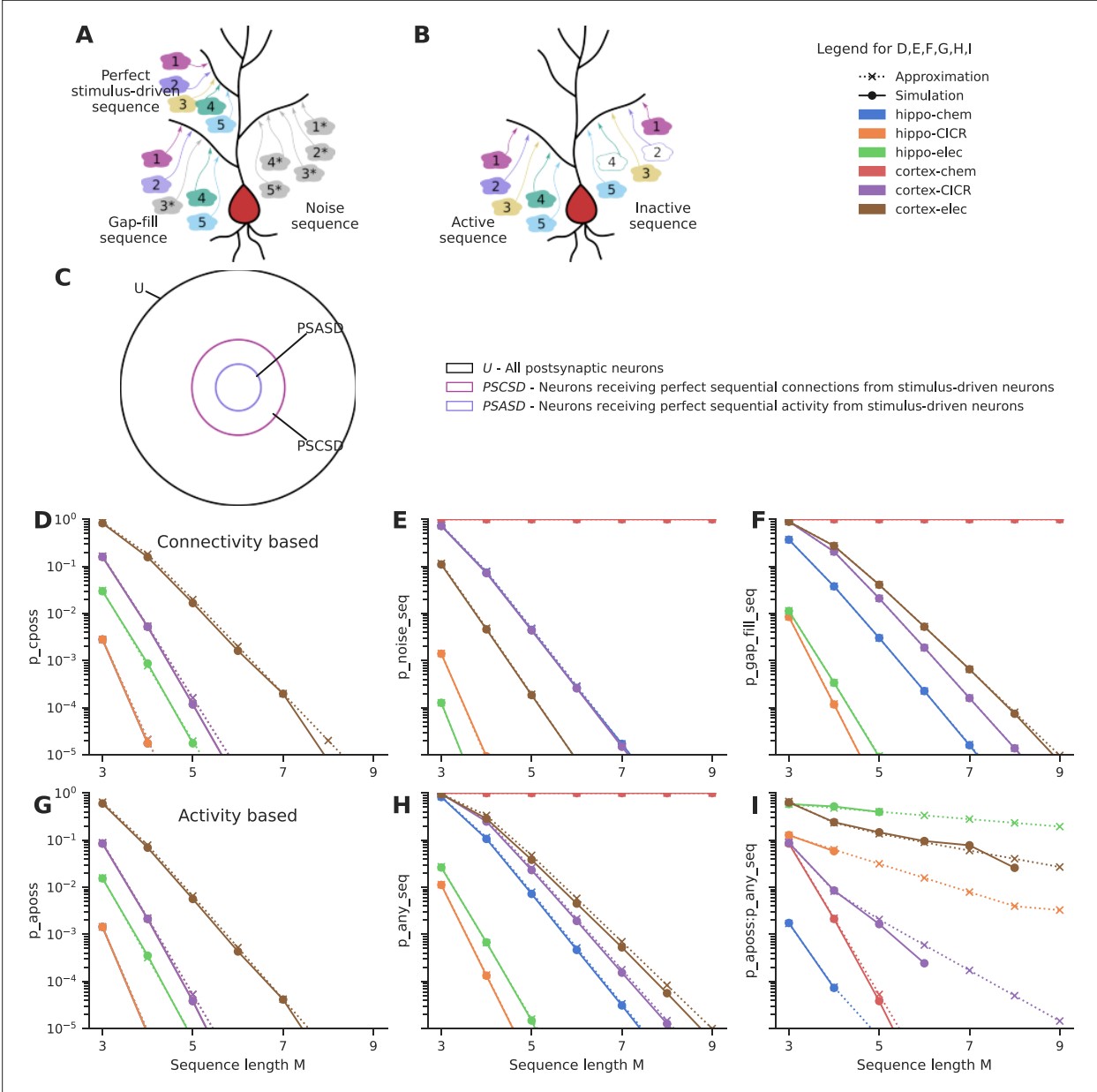

**Figure 4.** Sequential convergence of perfectly ordered stimulus sequences is likely in all network configurations in the presence of larger ensembles. (**A**) Different kinds of sequences. True sequences are formed of ensemble inputs alone, noise sequences are formed of background inputs while gap-fills consist of a combination of ensemble and background inputs. (**B**) Active and inactive sequences. An active sequence receives all constituting inputs, whereas an inactive sequence may be missing one or more inputs in the duration of occurrence of a sequence, in spite of receiving axonal connections from the ensembles. (**C**) Neurons classified as per the types of sequences they receive. Note that the schematic is a qualitative representation. The sizes of the circles do not correspond to the cardinality of the sets. (**D**) Probability of occurrence of connectivity-based perfectly ordered stimulus sequence which receives connections from all active ensembles in an ordered manner. The calcium-induced calcium release ('CICR') configuration overlaps with the 'chem' configuration of the corresponding network as they share the same values for S and Δ. (**E**) Probability of occurrence of noise sequences due to local noisy activation of synapses in an ordered fashion. (**F**) Probability of occurrence of gap-fill sequences that receive one or more, but not all inputs from ensembles. Background activity arriving at the right location time compensates for the inputs missing to make a perfect sequence. (**G**) Probability of occurrence of activity-based perfectly ordered stimulus sequence which receives inputs from all active ensembles in an ordered manner. (**H**) Probability of occurrence of a sequence due to the ordered convergence of either stimulus-driven inputs or noisy inputs or a combination of both. (**I**) Ratio of the probability of occurrence of active perfectly ordered stimulus sequences to the probability of occurrence of any sequence.

The online version of this article includes the following figure supplement(s) for figure 4:

**Figure supplement 1.** Factors affecting the likelihood of sequential convergence for different kinds of sequences.

**Figure supplement 2.** Probability of occurrence of small sequences that receive inputs from two ensembles.

Similar to the case with groups, noise sequences are more probable in the cortical configurations than hippocampal configurations owing to the higher rate of background activity (*Figure 4E*). Cortex-chemical is again the worst affected configuration owing to the larger time window for integration. Again, smaller noise sequences of three to five inputs are much more likely than longer ones. We find that gap-fill sequences constitute the major fraction of sequences in most of the network configurations (compare *Figure 4F, H*, *Figure 4—figure supplement 1F*). Hippo-elec again stands out as the configuration with the best signal-to-noise ratio (*Figure 4I*). Cortex-elec and hippo-CICR are the next couple of configurations that might be suitable for sequence detection.

The influence of various parameters on the probabilities of different kinds of sequences is examined in *Figure 4—figure supplement 1* using cortex-CICR as an example, varying one parameter at a time. Dense connectivity, higher participation probability of ensemble neurons, longer input zone width (Δ), and larger ensembles all increase the probability of occurrence of perfectly ordered stimulus sequences. As the ensembles get larger, gap-fills become major contributors of false-positive responses (*Figure 4—figure supplement 1F*). Gap-fills could play a dual role. Under low noise conditions, such as in the hippo-elec configuration, gap-fills could support sequence completion in the presence of partial input. However, under high noise conditions, such as in the cortex-chem configuration, their high likelihood of occurrence can lead to very dense representations in the population, thus hampering sequence discrimination.

Our results indicate that sequential convergence of three to five inputs is likely with ensembles of ~1000 neurons. Low noise conditions seem more suitable to the computation as gap-fills and noise sequences are less likely. Given the uncorrelated nature of background activity, noisy inputs arriving out of order (i.e. ectopic inputs) with respect to the sequence can influence the overall selectivity for the sequence. In the next sections we examined this question in two steps. First, using chemical and electrical models, we tested the effect of ectopic inputs arriving on or nearby an ongoing sequence. Next, we simulated neuronal sequence selectivity in a network including multiple patterns of ensemble stimulation, and noise, using an abstract formulation of sequence selectivity on dendrites.

## Ectopic inputs within the stimulus zone degrade selectivity for chemical sequences

A key concern with any estimate of selectivity in a network context is the arrival of extra inputs which interfere with sequence selectivity. We used computer models to estimate the effect of ectopic inputs on sequence selectivity. In this section we describe the outcomes on selectivity mediated by chemical mechanisms. We assume that the results apply both to biochemical signaling mechanisms and to CICR, using different timescales for the respective signaling steps.

We used a published sequence-selective reaction-diffusion system (*Bhalla, 2017*) involving a Ca-stimulated bistable-switch system with inhibitory feedback (molecule B) which slowly turns off the activated switch (molecule A) (*Figure 5A*). This system is highly selective for ordered/sequential stimuli in which the input arrives on successive locations spaced 3 μm apart along the simulated dendrite, at successive times (2 s intervals) as compared to scrambled input in which successive locations were not stimulated in sequential order (*Figure 5B and C*). We introduced ectopic input at points within as well as flanking the zone of regular sequential input (*Figure 5D*). We did this for three time points: at the start of the stimulus, in the middle, and at the end of the stimulus. In each case we assessed selectivity by comparing response to ordered input with response to a number of scrambled input patterns. Selectivity was calculated as

$$Selectivity = \frac{\left(A_{seq} - A_{mean}\right)}{A_{max}} \tag{6}$$

where $A_{seq}$ was the response to an ordered sequence, $A_{mean}$ was the mean of responses to all patterns, $A_{max}$ was the maximum response obtained among the set of patterns tested.

Our reference was the baseline selectivity to sequential input in the absence of ectopics. We observed that selectivity was strongly affected when ectopic input arrived within the regular stimulus zone, but when they were more than two diffusion length constants away they had minimal effect (*Figure 5E*) (D=5 and 2 μm$^2$/s for molecules A and B, respectively).

These calculations were for a sequence length of 5. We next asked how ectopic inputs impacted selectivity for sequence lengths of M=3–10. We restricted our calculations to within the stimulus zone

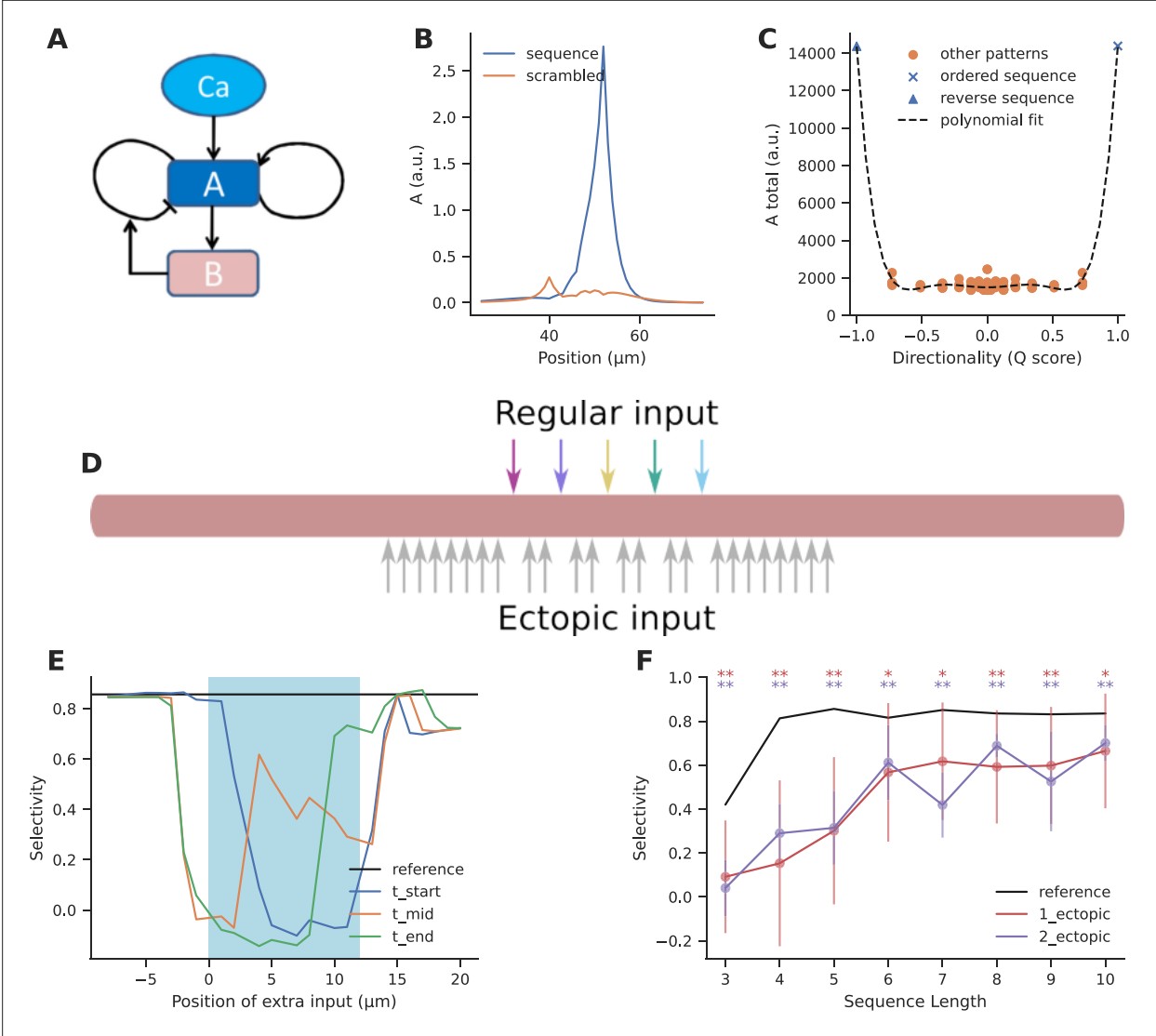

**Figure 5.** Ectopic input to chemical reaction-diffusion sequence selective system. (**A**) Schematic of chemical system. Calcium activates molecule A, which activates itself, and has negative inhibitory feedback involving molecule B. (**B**) Snapshot of concentration of molecule A in a one-dimensional diffusion system, following sequential (blue) and scrambled (orange) stimuli (methods). (**C**) Nonlinear response to sequential patterns. Total amount of molecule A integrated over space and time in the dendritic segment plotted against the directionality of the pattern. Black dashed line indicates the curve fit using a sixth order polynomial. (**D**) Geometry for analysis of influence of ectopic stimulation. Colored arrows represent the regular input for sequential/scrambled patterns. Gray arrows represent positions where the effect of ectopic inputs was tested. (**E**) Selectivity as a function of location of ectopic input. Ectopic input was given at three times: the start of the regular stimulus, the middle of the stimulus, and at the stimulus end. Cyan shaded region marks the zone of regular inputs. Ectopic stimuli were given at 1 μm spacing intervals to capture any steep changes in selectivity. The reference line indicates the baseline selectivity in the absence of any ectopic input. (**F**) Selectivity as a function of sequence length and number of ectopic inputs. Dots represent mean selectivity over different spatiotemporal combinations of the ectopic input/s and error bars represent standard deviation. Significance testing was done using Wilcoxon one-sample left-tailed test by comparing the 1-ectopic and 2-ectopic cases with the reference selectivity for each sequence length (p-values have been included in ***Supplementary file 1d***).

since this was where the impact of ectopic input was largest. We took the average of the selectivity for different spatiotemporal combinations of a single ectopic input within the stimulus zone, and compared it to the reference selectivity for the sequence without any other input (***Figure 5F***). We found that for M<5, selectivity was strongly affected (65–81% dip), but for long sequences (M ≥ 6), selectivity recovered substantially (20–30% dip). The reduction in selectivity due to two ectopic inputs was similar to that caused by a single ectopic (***Figure 5F***).

Overall, ectopic inputs severely degraded selectivity for short sequences but had a smaller effect on long ones. Selectivity was not strongly affected by ectopic inputs even as close as 3 µm away from the sequence zone.

## Ectopic inputs at the distal end degrade selectivity to a greater extent for electrical sequences

We next performed a similar analysis for electrical sequence selectivity, based on published experimental and modeling work from *Branco et al., 2010*. We adapted the Branco model of dendritic sequence selectivity to test the effect of ectopic input. We replicated the observation that synaptic inputs arriving in a distal-to-proximal (inward) manner at ~2.5 µm/ms elicited a higher depolarization than those arriving in a proximal-to-distal (outward) direction, or inputs that arrived in a scrambled order (*Figure 6B*). The selectivity we obtained is slightly lower than observed by *Branco et al., 2010*, since we tested a shorter sequence of five inputs as opposed to the longer sequence of length 9 that was used by the authors. Even though a 5-length sequence elicited relatively small EPSPs (4–6 mV) at the soma, the EPSPs elicited in the dendrite were sufficiently large (20–45 mV) to engage the NMDA-mediated nonlinearity mechanism (*Figure 6—figure supplement 2*). As a baseline, we obtained ~8–10% selectivity for sequences of length 5–9, where inputs arrived at a spacing of ~10 µm on a 99 µm long dendrite (*Figure 6D and E* - reference selectivity). Selectivity was maximum at input velocities between 2 and 4 µm/ms, similar to what was observed experimentally (*Branco et al., 2010*) for sequences of length 5–9 (*Figure 6C*). Another key distinction is that we measured selectivity as defined by *Equation 6* where A represented the peak potential recorded at the soma, whereas (*Branco et al., 2010*), referred to direction selectivity primarily as the difference between the responses to inward and outward input patterns.

We introduced ectopic synaptic input at different locations and times on the dendrite for ordered and scrambled sequences of five inputs (*Figure 6D*). Ectopic inputs both within and outside the zone degraded selectivity. Selectivity was most strongly disrupted by ectopic inputs arriving at the distal end. Due to the high input impedance at the distal end, ectopic inputs arriving there were not just amplifying the inward sequence response, but were boosting responses to scrambled patterns as well. The resulting narrower distribution of peak somatic EPSPs led to a further drop in selectivity (*Figure 6—figure supplement 1*).

We further tested the effect of ectopic inputs on sequences of different lengths. Ectopic inputs were delivered near the start, middle, and end of the sequence zone, at three different times coinciding with the start, middle, and end of the sequence duration. Short sequences of three to four inputs had low baseline selectivity, which was slightly amplified upon addition of ectopic input. With mid-length sequences of five to seven inputs, selectivity dropped by 27–53%. Longer sequences were less affected by ectopics (16–18% drop) (*Figure 6E*).

In summary, ectopic inputs degraded electrical sequence selectivity even in the flanking regions, particularly when they were delivered closer to the distal end of the dendrite. Again, longer sequences were less perturbed than mid-length ones.

## Strong selectivity for sequences at the dendrite translates to weak selectivity at the soma

Neurons receive a wide range of inputs across their dendritic arbors. Since our previous simulations indeed showed that ectopic inputs can affect sequence selectivity, we wanted to assess how selective neurons are for sequential activity occurring on a short segment of dendrite, in the midst of all the other activity that they receive, particularly in a network scenario. In the context of a randomly connected feedforward network, can neurons receiving *perfect sequential connectivity from stimulus-driven neurons* (PSCSD neurons) be distinguished from other neurons based on their selectivity for sequential inputs? Since chemical sequences showed stronger reference selectivity (~0.8) when compared to the electrical case (~0.1) and owing to the low noise conditions, we chose the hippo-CICR configuration with four sequentially active ensembles to estimate sequence selectivity at the neuron level.

To test this, we devised an abstract formulation to calculate a neuron's activation based on all inputs it receives. The abstract formulation made the analysis scalable for a network context. At each location on the dendrite, the local increase in activation was a function of two things: the baseline

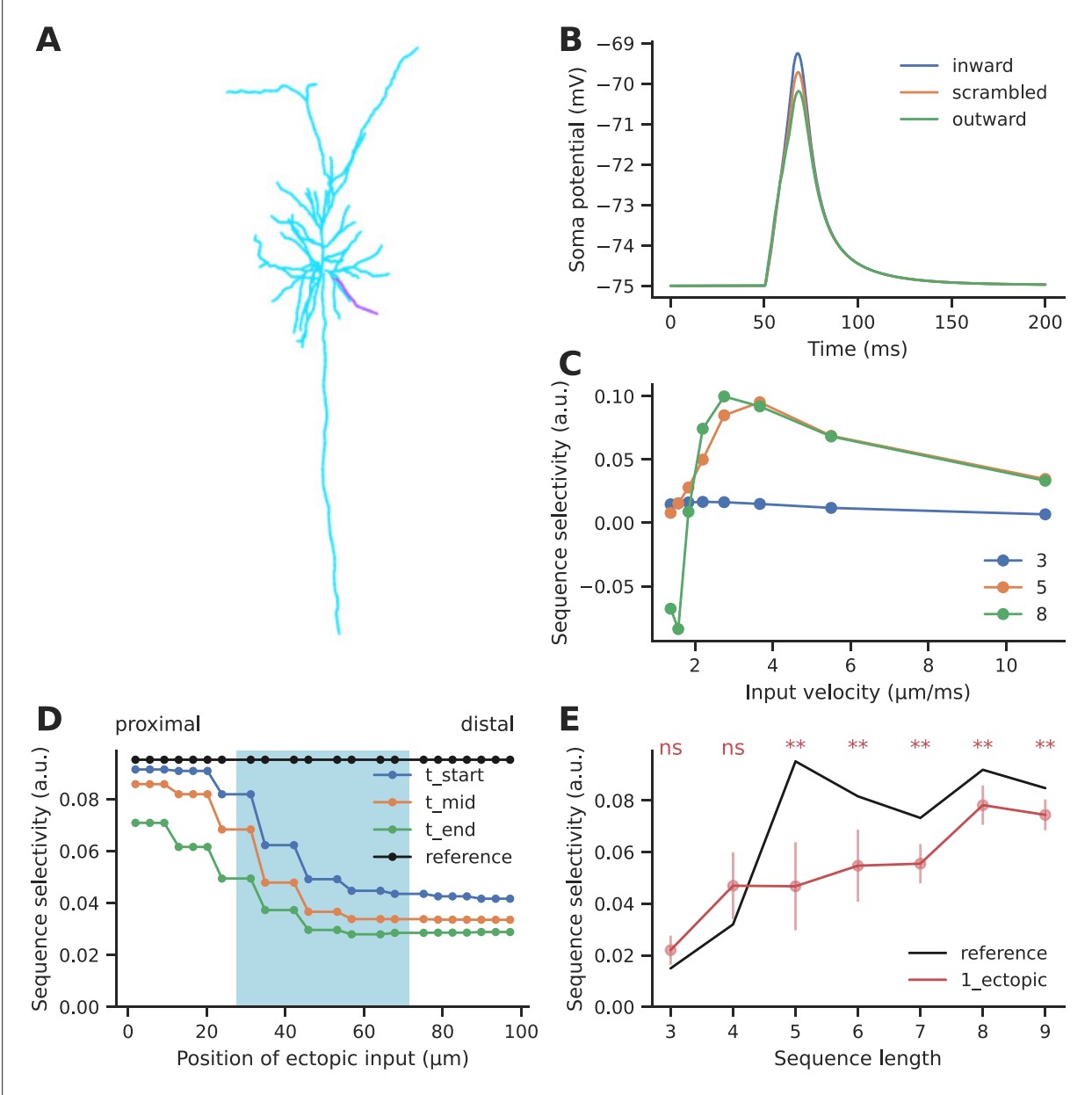

**Figure 6.** Effect of ectopic input on electrical sequence selectivity. (**A**) Neuronal morphology used for examining the effect of ectopic input in an electrical model. The selected dendrite for examining selectivity has been indicated in magenta. (**B**) Soma response to three different input patterns containing five inputs: perfectly ordered inward sequence, outward sequence, and scrambled pattern. (**C**) Dependence of selectivity on stimulus speed, i.e., how fast successive inputs were delivered for a sequence of five inputs. (**D**) Effect of ectopic input at different stimulus locations along the dendrite, for sequence of length 5. Shaded cyan region indicates the location of the regular synaptic input. Stimuli were given at time t_start: the start of the regular stimulus sequence; t_mid: the middle of the sequence, and t_end: the end of the sequence. (**E**) Effect of single ectopic input as a function of sequence length. Selectivity drops when there is a single ectopic input for sequences with ≥ 5 inputs. Dots represent mean selectivity over different spatiotemporal combinations of the ectopic input and error bars represent standard deviation. Significance testing was done using Wilcoxon one-sample left-tailed test by comparing the 1-ectopic case with the reference selectivity for each sequence length (p-values have been included in *Supplementary file 1e*).

The online version of this article includes the following figure supplement(s) for figure 6:

**Figure supplement 1.** Strong disruption of selectivity at the distal end is caused by decrease in variance among the responses to different patterns.

**Figure supplement 2.** NMDAR activation occurs even for sequences of five inputs.

activation contributed by an input arriving at that location and the activation buildup due to sequential inputs (Methods). The sequential contribution was scaled nonlinearly using an exponent and passed through a shifted sigmoid that bounded the maximum local activation attained within a time step. Hence, sequential inputs that follow the S to S+Δ spacing rule resulted in large local activations. The local activations were integrated over time and space to obtain the overall activation of a neuron. This process was repeated for different stimulus patterns where the ensembles were activated in a different order in each pattern.

Neurons receiving sequential connectivity for the ordered stimulus [1,2,3,4] (PSCSD neurons) showed slightly higher activations on average relative to neurons that did not receive such connectivity (PSCSD[C] neurons) (*Figure 7A, D, and E*). PSCSD neurons also showed higher activation to sequential stimuli than to scrambled stimuli, e.g., [3,1,2,4] (compare reds in *Figure 7A* with B, F with G) or reverse stimuli [4,3,2,1] (compare reds in *Figure 7A* with C, F with H). PSCSD neurons showed slightly higher selectivity for the ordered sequence (*Figure 7J*) than PSCSD[C] neurons (*Figure 7K*).

Despite using a model that was tuned to show high sequence selectivity (~0.8) at the dendrite, we observed weak somatic selectivity in the range of 0.05–0.0.08 (mean = 0.06, std = 0.01), when we accounted for all the inputs they received across their arbor. This was because the total activation of background activity and other non-sequential inputs over the entire dendritic tree overwhelmed the small signal elicited by the occurrence of a sequence. Specifically, each neuron receives ~160 inputs from background activity over the duration of the sequence. In the presence of ensemble activity, it receives an additional ~200 inputs scattered over the dendritic arbor. Hence, a sequence of four inputs has a relatively small effect on this high baseline, even when it is scaled by strong nonlinearities at the local dendritic zone encompassing all the inputs in the sequence. We postulated that this problem can be alleviated when the neurons are in a balanced state, where the excitation is precisely balanced with inhibition (*Hennequin et al., 2017*; *Bhatia et al., 2019*). A previous model for sequence selectivity (*Bhalla, 2017*) included inhibition. To nullify the effect of background activity as a proxy for the EI balanced state, we subtracted the trial-averaged activation under conditions of background activity from the trial-averaged activation value seen when ensembles are stimulated in addition to the background activity. This improved the selectivity to the range 0.13–0.22 (mean = 0.17, std = 0.02).

As a comparison, we ran the same simulation for the cortex-CICR configuration with 4× denser connectivity and 10× higher background activity. Under these high noise conditions, even PSCSD[C] show frequent occurrences of false positive (noise/gap-fill sequences). Further, after subtracting the baseline activation levels observed with background activity alone, the PSCSD neurons showed poor selectivity in the range of –0.03 to 0.07 (mean = 0.02, std = 0.02) (*Figure 7—figure supplement 1*). Hence, sequence selectivity operates best under conditions of low noise.

Overall, in this section we find that even strong dendritic selectivity may yield only small somatic responses which may require additional constraints such as EI balance and low background activity in order to be resolved at the soma.

## A single model exhibits multiple forms of nonlinear dendritic selectivity

We implemented all three forms of selectivity described above, in a single model which included six voltage and calcium-gated ion channels, NMDA, AMPA, and GABA receptors, and chemical signaling processes in spines and dendrites. The goal of this was threefold: To show how these nonlinear operations emerge in a mechanistically detailed model, to show that they can coexist, and to show that they are separated in timescales. We implemented a Y-branched neuron model with additional electrical compartments for the dendritic spines (Methods). This model was closely based on a published detailed chemical-electrical model (*Bhalla, 2017*). We stimulated this model with synaptic input corresponding to the three kinds of spatiotemporal patterns described in figures: *Figure 8—figure supplement 1* (sequential synaptic activity triggering electrical sequence selectivity), *Figure 8—figure supplement 2* (spatially grouped synaptic stimuli leading to local Ca4_CaM activation), and *Figure 8—figure supplement 3* (sequential bursts of synaptic activity triggering chemical sequence selectivity). We found that each of these mechanisms show nonlinear selectivity with respect to both synaptic spacing and synaptic weights. Further, these forms of selectivity coexist in the composite model (*Figure 8—figure supplements 1–3*), separated by the timescales of the stimulus patterns (~100 ms, ~1 s, and ~10 s, respectively). Thus, mixed signaling in active nonlinear dendrites yields selectivity of the same form as we explored in simpler individual models. A more complete analysis of

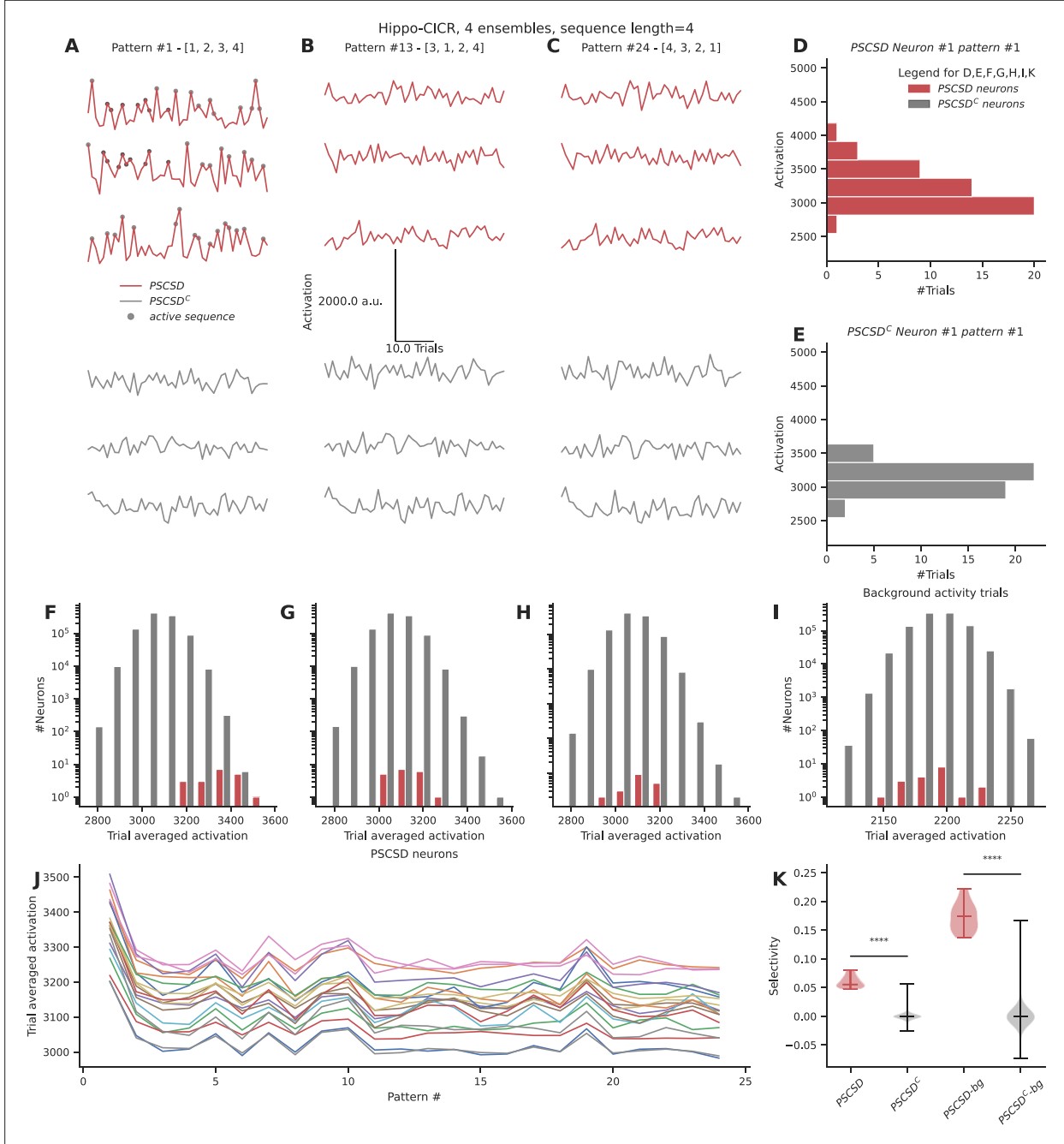

**Figure 7.** Neurons receiving perfectly ordered stimulus sequences show weak selectivity. (**A, B, C**) Representative total neuronal activation across 48 trials of three neurons that receive perfect sequential connections from stimulus-driven neurons (PSCSD) (red) and three neurons that do not receive such connections from stimulus-driven neurons (PSCSD$^C$) (gray) for three stimulus patterns: Perfect ordered activation of ensembles (**A**), scrambled activation (**B**), and activation of ensembles in the reverse order (**C**). Gray dots mark the occurrence of any sequence. (**D, E**) Representative activation values across trials for pattern #1 for the first neuron in each group: the PSCSD group shown in red (**D**) and the PSCSD$^C$ group shown in gray (**E**). Neurons are riding on high baseline activation, with small fluctuations when a complete active sequence occurs. (**F, G, H**) Trial-averaged neuronal activation of PSCSD vs PSCSD$^C$ neurons, for the three stimulus patterns indicated in **A, B, C** respectively. (**I**) Trial-averaged neuronal activation of neurons classified as PSCSD vs PSCSD$^C$ in the presence of background activity alone, i.e., when the ensembles are not stimulated. (**J**) Trial-averaged neuronal activation of 19 PSCSD neurons (depicted in different colors) in response to 24 different stimulus patterns. Pattern #1 corresponds to the ordered sequence, while pattern #24 is the reverse sequence. (**K**) Sequence selectivity of PSCSD and PSCSD$^C$ neurons, without (one-sided Mann-Whitney U test, p=2.18*10$^{-14}$) and with background activity subtraction (one-sided Mann-Whitney U test, p=2.18*10$^{-14}$).

The online version of this article includes the following figure supplement(s) for figure 7:

*Figure 7 continued on next page*

*Figure 7 continued*

**Figure supplement 1.** Neurons receiving perfect stimulus-driven sequences show poor selectivity in the presence of high background activity, such as the cortex-calcium-induced calcium release (CICR) configuration.

the effect of morphology, branching, and channel distributions deserves a separate in-depth analysis, and is outside the scope of the current study.

## Discussion

We have shown using cortical and hippocampal statistics that random feedforward connectivity, along with large enough ensembles (~100 neurons for groups and ~1000 neurons for sequences), is sufficient to lead to the convergence of groups or sequences consisting of three to five inputs. This does not require further mechanisms for activity or proximity-driven axonal targeting in a population of ~100,000 postsynaptic neurons. Such convergence provides a substrate for downstream networks to decode arbitrary input patterns.

### Testing connectivity predictions using structural connectivity

Considerable advances have been made in establishing structural connectivity based on EM reconstruction (*Takemura et al., 2013*; *Lee et al., 2016a*; *Zheng et al., 2018*), viral projection-labeling (*Ugolini, 1995*; *Wickersham et al., 2007*; *Wall et al., 2010*; *Eastwood et al., 2019*; *Oh et al., 2014*), and RNA barcoding (*Chen et al., 2019*; *Sun et al., 2021*) approaches. There have been numerous critiques of such approaches in understanding network computation (*Bargmann and Marder, 2013*; *Gomez-Marin, 2021*). Our analysis bridges such structural observations of connectivity and their functional implications. We focus on connectivity and activity of dendritic clusters as a potential computational motif (*Pulikkottil et al., 2021*). This focus is supported both by the known occurrence of such connectivity including functional readouts (*Ju et al., 2020*; *Adoff et al., 2021*; *Kerlin et al., 2019*; *Fu et al., 2012*) and by postsynaptic mechanisms for detecting such clustered activity (*Gasparini and Magee, 2006*; *Branco et al., 2010*).

Our first-pass analysis shows that connectivity leading to fully mixed conventional 'grouped' synaptic clusters of three to four inputs is likely in feedforward networks with random connectivity (*Figure 2F*). Shorter groups are so dense as to occur on most neurons, which makes them hard to decode (*Foldiak, 2003*). Further, if the uniqueness constraint is relaxed, stimulus-driven groups that receive three to five convergent connections from any ensemble neurons are up to two orders of magnitude more likely than fully mixed groups (*Figure 2G*). Due to the additional constraint imposed by order, sequential convergence required greater redundancy in the input representations in the form of larger ensembles to give rise to similar probabilities of convergence (*Figure 4*).

The olfactory bulb mitral cell projections to pyramidal neurons in the piriform cortex (PC) are an example where anatomy provides predefined ensembles in the form of glomeruli, and projections that are almost random (*Ghosh et al., 2011*). We used our analysis to obtain insights into this circuit. In this calculation, an input ensemble consists of mitral cells receiving input from a single class of olfactory receptor neurons (ORNs). Each glomerulus is innervated by the apical tuft by ~70 mitral/tufted cells, of which about 20 are mitral cells which project to piriform (*Nagayama et al., 2014*). Since each ORN class projects to two glomeruli (*Ressler et al., 1994*), ~40 mitral cells are co-activated upon activation of a specific receptor class, thus forming a structurally and functionally well-defined ensemble. When odors arrive, olfactory bulb glomeruli show similar sequential activity profiles as depicted in *Figure 1A* (*Spors and Grinvald, 2002*; *Schaefer and Margrie, 2007*; *Junek et al., 2010*). Mitral cells predominantly project to the apical dendrites of PC neurons. These connections appear to be distributed and lack spatially structured organization within the PC (*Ghosh et al., 2011*; *Sosulski et al., 2011*; *Miyamichi et al., 2011*; *Stettler and Axel, 2009*; *Illig and Haberly, 2003*), and therefore, we treat them as randomly connected.

In mice, each PC neuron, on average, makes 0.64 synapses with neurons from a single glomerulus (*Srinivasan and Stevens, 2018*). Thus, our estimate for the expected number of inputs from a single ensemble, i.e., p*N is ~1.28 (=2*0.64). With an apical dendritic arbor of ~2000 μm (*Moreno-Velasquez et al., 2020*), in a population of ~500,000 PC neurons present in one hemisphere (*Srinivasan and Stevens, 2018*), we predict that sparse representations of grouped mitral cell projections from three

to four pairs of glomeruli are likely over dendritic zones ~50 μm ($P_{cFMG}$~3.93*10$^{-5}$, $P_{cSDG}$~4.04*10$^{-4}$ for group size of 4). However, convergence over the CICR/chemical length scales of 10 μm is less probable ($P_{cFMG}$~3.31*10$^{-7}$, $P_{cSDG}$~3.51*10$^{-6}$ for group size of 4). κ is considered to be L/Z for these calculations. Sequential convergence over spacing windows of 10 μm is even less likely ($P_{cPOSS}$~4.19*10$^{-8}$). Hence, PC neurons are likely to receive grouped projections from three to four different pairs of glomeruli (representing a single odor) onto individual dendritic branches even with random connectivity. These grouped inputs could trigger NMDA spikes in the apical tuft dendrites (*Kumar et al., 2018*) which could aid in discriminating between odors and mediating plasticity for odor memory formation (*Kumar et al., 2021*).

Based on these calculations, we predict that optogenetic stimulation of combinations of three to four glomeruli should (a) show clustered glutamate release on PC dendrites (can be validated using iGluSnFR [*Marvin et al., 2013*; *Marvin et al., 2018*]), and (b) show nonlinear summation as recorded from PC dendrites and possibly somas.

A similar analysis may be applicable to other neuronal circuits such as the barrel cortex and other somatosensory regions as places where the anatomy may lend itself to following up such a connectivity analysis. However, our analysis is general and applies to any circuit where computation-specific ensembles arise and connectivity is approximately random, even if they lack the convenient anatomical demarcation of ensembles provided by the olfactory bulb or somatosensory regions.

## Random connectivity in conjunction with strong dendritic nonlinearities supports mixed selectivity of three to four different inputs

Mixed selectivity, wherein neurons are selective to combinations of task variables as opposed to single variables, has gained a lot of interest recently (*Rigotti et al., 2013*; *Fusi et al., 2016*; *Eichenbaum, 2018*). Neurons showing mixed selectivity have been observed in the vibrissal cortex (*Ranganathan et al., 2018*), hippocampus (*Stefanini et al., 2020*), prefrontal cortex (*Kobak et al., 2016*) (based on *Romo et al., 1999*; *Brody et al., 2003*) among other brain regions. Nonlinear mixed selectivity, wherein a neuron's activity is not a simple weighted linear combination of the different stimuli, has the potential to expand the dimensionality of neural representations (*Fusi et al., 2016*). Such high-dimensional representations are particularly helpful in tasks involving the mixing of multiple stimuli/variables because they enable simple downstream linear readout mechanisms to decode patterns.

How do neurons achieve mixed selectivity? A possible avenue is through dendritic nonlinearities. For example, work from *Ranganathan et al., 2018*, in the vibrissal cortex has shown that mixed selectivity can arise from active dendritic integration in the layer 5 pyramidal neurons, which show joint representation for sensorimotor variables. This is particularly useful in the context of adaptive sensing. Our study explores this possibility of whether dendritic computation could act as a mechanism for neurons to achieve mixed selectivity. We show that redundancy in input representation in the form of large presynaptic ensembles yields postsynaptic neurons showing different kinds of selectivity: including pure selectivity, partially mixed and fully mixed selectivity. In fact, it is far more likely to find neurons showing partially mixed selectivity than neurons showing pure/fully mixed selectivity. While such partial mixing makes it difficult to decode the occurrence of full patterns, they may still act as building blocks for neurons further downstream to aid in the decoding process.

We show that under conditions of low noise, random connectivity coupled with strong dendritic nonlinearities provides an excellent substrate for neurons to mix different sources of information. Stimulus-driven groups of three to five inputs are likely with ensembles of 100 neurons (*Figure 2G*) in four of the six network configurations tested. This mixing occurs over short segments of dendrites, enabling each neuron to potentially mix multiple different combinations of three to five inputs. Since partially mixed groups are more likely than fully mixed groups, decoding the identity of patterns would require further processing based on the population response of all such neurons.

## Effective pattern representation through dendritic computation needs more than structural and functional connectivity

A key prediction of our study is that structural and even functional connectivity are necessary but not sufficient conditions to achieve postsynaptic computation in clusters and sequences. Several additional factors influence the effective selectivity for clustered/sequential computation. Our analysis of the hippocampal and cortical configurations occurring over three spatiotemporal scales (electrical,

CICR, and chemical) indicates that background activity plays a pivotal role in determining the ability of the network to extract useful information from activity patterns. Hence, we found that the hippo-campal configurations are better suited for grouped as well as sequential computations, as compared to the cortex (*Figures 2 and 4*).

Our analysis sets boundary conditions on likely postsynaptic mechanisms for dendritic cluster computations. Specifically, chemical computations occurring over timescales of several seconds are vulnerable to background inputs whose likelihood of hitting a synapse scales with the duration of the computation. (*Figure 2H and I*). This effect is even more pronounced in the cortex due to its higher background activity (*Buzsáki and Mizuseki, 2014*). Hence, we posit that chemical timescales are less useful for ongoing activity-driven computation leading to changes in cell firing, and rather more useful from the context of plasticity. Chemical processes occurring over seconds to minutes can modulate synaptic weights locally within dendritic clusters, strengthening grouped inputs through cooperative plasticity (*Harvey et al., 2008*). Similarly, our analysis showed that electrical mechanisms worked well for grouped computations, but they were quite sensitive to ectopic inputs in the case of sequences. Further, some studies suggest that three to five inputs may be insufficient to trigger dendritic spikes (*Gasparini et al., 2004*), although *Goetz et al., 2021*, argue that a few very strong synapses can also trigger them. CICR operating on the timescales of ~100–1000 ms, on the other hand, could be a potential mechanism that could influence both computation and plasticity. Higher level of dendritic $Ca^{2+}$ affects activity at the soma through dendritic spikes (*Palmer et al., 2014*) and enhances dendrite-soma correlations (*O'Hare et al., 2022*). In addition, $Ca^{2+}$ is known to trigger downstream mechanisms that can modulate plasticity (*Mateos-Aparicio, 2020*). Hence, CICR is a promising candidate that could boost the impact of dendritic computations in a network context.

Sequence selectivity required low background activity and strong nonlinearities to amplify sequential inputs over other kinds of inputs at the dendritic level. Chemical mechanisms showed stronger selectivity than electrical mechanisms, and they were less prone to ectopic inputs arriving outside the zone of the sequence (*Figures 5 and 6*). However, despite using strong nonlinearities at the dendritic zone, neurons receiving perfect sequential connectivity (PSCSD) showed weak selectivity at the somatic level when we accounted for inputs arriving across the entire dendritic arbor. The sheer number of background or non-sequential ensemble inputs impinging on the dendritic arbor led to elevated baseline activation. This problem could be alleviated through precise EI balance. If the neurons exist in the state of precise EI balance (*Hennequin et al., 2017*; *Bhatia et al., 2019*), inhibition could cancel out the effect of background activity thus amplifying the contribution of sequential inputs to the neuron's activation.

Thus, strong nonlinearities, timescales of a few 100 ms, and low noise regimes are best suited for both grouped and sequence computations. Precise EI balance in the background inputs improves the conversion of dendritic sequence selectivity to somatic sequence selectivity.

Overall, our study provides a general framework for assessing when dendritic grouped or sequence computation can discriminate between different patterns of ensemble activations in a randomly connected feedforward network. Our approach may need further detailing to account for area-specific features of connectivity, synaptic weights, neuronal morphologies, dendritic branching patterns, and network architectures. Conversely, deviations from our theory may suggest where to look for additional factors that determine connectivity and neuronal selectivity. Our analysis touches lightly on the effects of inhibition and plasticity, though these too may contribute strongly to the outcome of clustered input. While connectivity is a necessary substrate for clustered dendritic computations, we show that factors such as background activity, length, and timescales of postsynaptic mechanisms, and dendritic nonlinearities can play a key role in enabling the network to tap into such computations. In addition, our analysis can guide the design of biologically inspired networks of neurons with dendrites for solving a variety of interesting problems (*Chavlis and Poirazi, 2024*; *Iyer et al., 2022*; *Asabuki and Fukai, 2020*; *Asabuki et al., 2022*).

## Methods

The derivations for the equations have been included in the main text and in Appendix 1. All simulations used Python version 3.8.10. We used both analytical equations and connectivity-based simulations for the results shown in *Figures 2 and 4*. Simulations were run using a 128-core machine running Ubuntu server 22.04.2 LTS. All code used for this paper is available at GitHub (https://github.com/

BhallaLab/synaptic-convergence-paper, copy archived at *Somashekar, 2025*). The readme file in the GitHub repository contains instructions for running the simulations.

For simulations shown in *Figures 2B, 5, and 8*, and their corresponding figure supplement, the MOOSE Simulator (*Ray and Bhalla, 2008*) version 4.0.0 was used. The NEURON simulator was used for *Figure 6*. Version numbers of all the Python packages used are available in the requirements.txt file on the GitHub repository mentioned above.

## Connectivity-based simulations

For both sequences and groups, we considered a feedforward network containing a presynaptic population of $T_{pre}$ neurons. Connectivity to neurons in the postsynaptic population was simulated by sampling inputs randomly from the presynaptic population onto a single long dendrite of length L, which was discretized into $p*T_{pre}(=20,000)$ uniformly distributed synapses. Here, p is the connection probability between the two layers. In each run connectivity is sampled for 4000 neurons. Hence, the connectivity to the dendrite was represented as a matrix of 4000 neurons × 20,000 inputs. As simulating connectivity and examining groups/sequences for the whole postsynaptic population in a serial manner was computationally time-consuming, we parallelized the process by breaking down the simulation into 100 runs consisting of 4000 postsynaptic neurons per run. This does not affect the results since the connectivity to each neuron is sampled independently, and the presynaptic activity was kept the same across different runs by initializing the same seed for generating it. However, since we wanted neurons with different connectivity, the seed for generating connectivity was changed in each run. Hence, we sampled connectivity for a total of 400,000 postsynaptic neurons in all simulations except for *Figure 7*, where we sampled 1,000,000 neurons.

## Generating presynaptic activity

The first N*M neurons in the presynaptic population represented the activity of M ensembles consisting of N neurons each. The remaining neurons participated in background activity, showing Poisson firing. A total of 100 trials, consisting of three trial types (connectivity-based, stimulus trials, and background-only trials as defined below), were simulated. In the connectivity-based trial, the ensemble participation probability, $p_e$, is set to 1, and there is no background activity. A single trial of this type was run to estimate connectivity-based groups/sequences. In stimulus trials, $p_e$ was set to 0.8 and the non-ensemble neurons are considered to be firing with a rate R in a Poisson manner. Background-only trials had no ensemble activity. All neurons in background-only trials were considered to be participating in background activity with a firing rate R. With the Poisson assumption, the probability of a neuron firing in duration D due to background activity was considered to be $p_{bg} = 1-e^{-RD}$. In each trial, the activity of each presynaptic neuron was sampled from a Bernoulli distribution with $p=p_e$ or $p=p_{bg}$ depending on whether it was an ensemble neuron or was participating in background activity. The activity of every neuron in the presynaptic (input) population was represented as a one, minus one, or zero, where ones specified that the neuron participated in ensemble activity, minus ones represented background activity, and zeros represented no activity.

## Estimating the probability of occurrence of groups

In the case of groups, neurons in all M ensembles were co-active within duration D with a probability of $p_e = 0.8$ in stimulus trials and $p_e = 1$ in connectivity-based trials. For each postsynaptic neuron an input array was generated where the entry at index i represented the ensemble ID of input arriving at position i on the dendrite if the input belonged to one of the ensembles. It was –1 if the input was due to background activity, and 0 if the synapse was inactive. The input array is scanned for the occurrences of groups by sliding a window of length Z, one synapse at time. For a group of size M, if there was at least one input from each of the M ensembles within zone Z, it was considered a *fully mixed group*. A *stimulus-driven group* was defined as one that received M or more inputs in zone Z from neurons belonging to any of the ensembles. If there were M or more inputs from background activity in a window of length Z, it was considered a *noise group*. We defined *any* group as one consisting of M or more inputs, either from the ensembles or from background activity or a combination of the two, arriving in a zone of length Z. The sum of the number of groups occurring on the dendrite was calculated for each group type (true, stimulus-driven, noise, and any). The probability of a group occurring

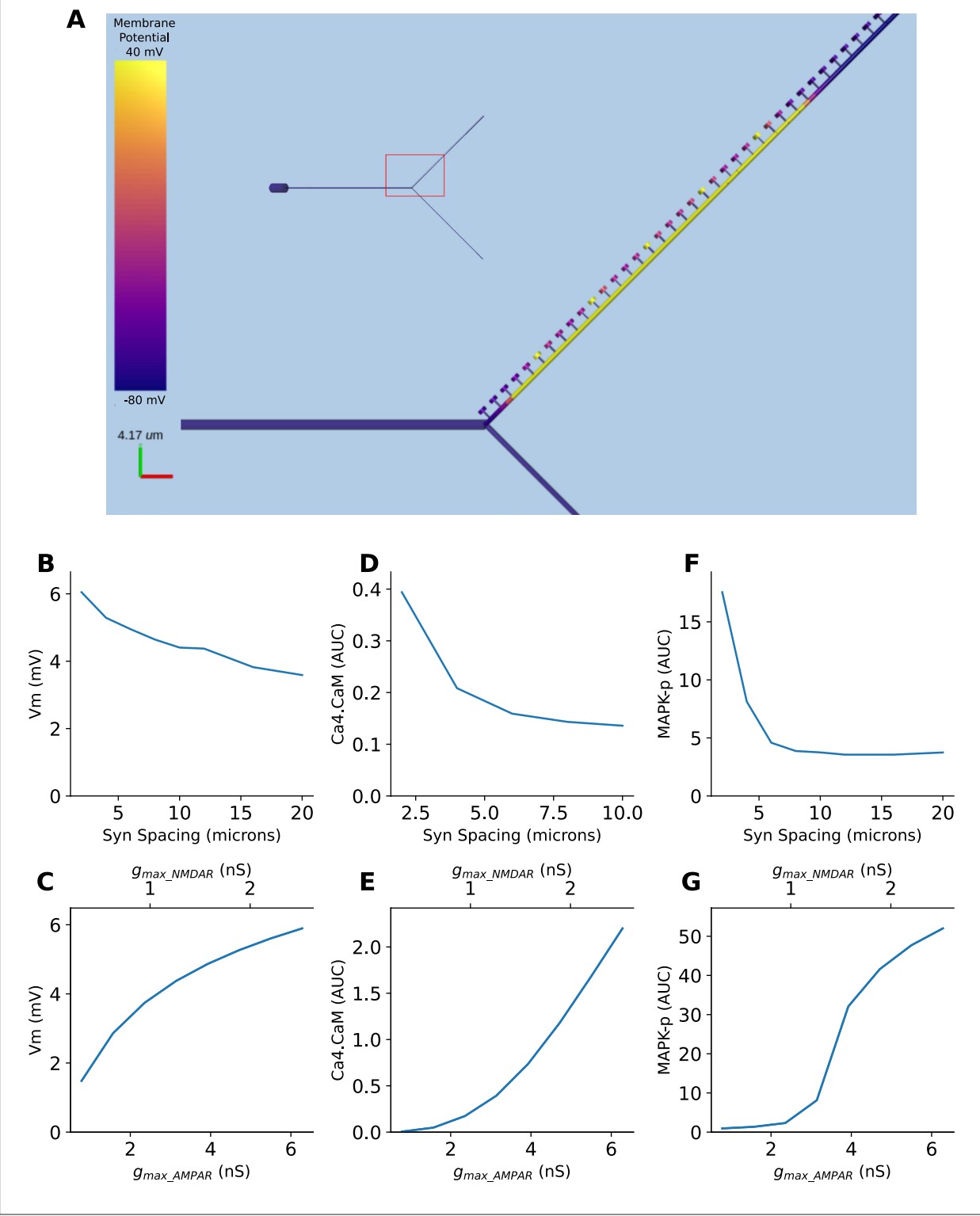

**Figure 8.** Multiple forms of nonlinear dendritic selectivity in a single neuron. (**A**) Model of Y-branched neuron with dendritic spines. The inset shows the morphology of the full neuron with the zoomed-in section highlighted in red. (**B, D, F**) Nonlinearity in electrical (Vm), Ca4-CaM, and MAPK-p responses as a function of synaptic spacing. (**C, E, G**) Nonlinearity in electrical (Vm), Ca4-CaM, and MAPK-p responses as a function of synaptic weight, expressed as AMPAR and NMDAR maximal conductances.

The online version of this article includes the following figure supplement(s) for figure 8:

**Figure supplement 1.** Response of multiscale model to inputs at electrical timescales.

*Figure 8 continued on next page*

*Figure 8 continued*

**Figure supplement 2.** Response of multiscale model to grouped and dispersed inputs arriving over the timescales of calcium and CaM dynamics.

**Figure supplement 3.** Response of multiscale model to inputs arriving at chemical timescales.

on a neuron was estimated by calculating the mean number of neurons in the postsynaptic population that received at least one group of that specific type somewhere along the dendrite.

## Estimating neuronal activation for the context of grouped computation

To explore the effect of nonlinear grouped dendritic computation on neuronal response, we applied a nonlinear function on local dendritic inputs. Using the input array generated in the previous step for detecting groups, the stretched exponential function depicted in the inset in *Figure 3A* was applied to the total number of inputs arriving on the dendrite within a zone of length Z. In this manner, using a rolling window approach the local dendritic activation at each location on the dendrite was estimated as the output of this nonlinearity. Finally, the net activation of the neuron was measured as the summed activation across the whole dendritic arbor of length L, normalized by the total number of synapses in the zone of length Z.

$$\text{Neuronal activation} = \frac{\sigma}{Z} \sum_{i=0}^{i=\frac{L}{\sigma}} \omega \left( \sum_{x=i}^{x=i+\frac{Z}{\sigma}} \delta\left(x\right) \right)$$

where $\omega$ represents the nonlinear function, $\delta(x)=1$ if an input arrived at location x, else it is 0. L is the total dendritic length, Z the zone length, and $\sigma$ the inter-synapse spacing.

The nonlinearity adopted here was inspired from the cumulative distribution function of the Weibull distribution. It has the following useful properties:

1. It is bounded, i.e., the local activation always saturates at 1.
2. It is nonlinear.
3. By controlling the exponent, one can control the effect fewer inputs have on the activation. This is particularly relevant for tuning the contribution smaller groups have on the activation of the neuron.
4. One can also control the fixed value of this distribution, which is the point that remains frozen upon changing powers. We have used a value of 10 as the base of the second term, which implies that when all M inputs constituting a group are present the resulting local activation is 0.9, i.e., 90% of the maximum. When there are >M inputs, the activation saturates at a value of 1.

## Estimating the probability of occurrence of sequences

A key difference between groups and sequences was that the presynaptic activity was simulated for a single time step in the case of groups. However, it was simulated over MAX_M (=9) time steps which was the maximum sequence length we considered. Hence, the presynaptic activity for sequences was represented as a matrix of ones, minus ones, and zeros of dimensions $T_{pre} \times 9$, where ones represented ensemble activity, minus ones represented background activity, and zeros represented no activity. In stimulus trials, at any given time step, only one ensemble was considered to be active. Neurons belonging to inactive ensembles in a particular time step were considered to participate in background activity in that time step. Although the presynaptic activity was simulated for a total of MAX_M time steps, while counting the number of sequences for a sequence of length M, only the activity from the first M time steps was used, as that was the length of the sequence we were interested in. Connectivity was simulated between the pre- and postsynaptic neurons as described earlier.

Using this setup, we looked for the occurrence of different kinds of sequences. Sequences consisted of inputs that occurred within a spacing of S to S+Δ from an input that arrived at a previous time step. *Perfectly-ordered stimulus sequences (POSS)* were made of ones alone. *Noise sequences* consisted of minus ones alone. *Any sequences* were formed from either ones or minus ones or a combination of both.

Since the pairwise relationships between inputs occurring in adjacent time steps was sufficient to capture whether two inputs followed the S to S+Δ spacing, identifying synapses that followed this rule across adjacent time steps aids in determining whether they participated in a sequence. Hence, we constructed distance matrices between synapses that followed the above spacing rule for adjacent time windows wherein the $X_{t\text{->}(t+1)}[i,j]$ entry in the matrix was 1 if synapse j received input in the time window t+1, following the arrival of input at synapse i in time window t, provided synapse j occurred within a spacing of S to S+Δ of synapse i. The total number of sequences consisting of M inputs occurring between different pairs of synapses was obtained by multiplying all pairwise distance matrices and summing the entries in the product.

Total number of sequences that occurred on a neuron = $\sum_{i,j} \prod_{t=1}^{t=M-1} X_{t \to (t+1)}$.

Distance matrices were constructed separately for the cases of true, noise, and all sequences for each neuron and the number of different sequences in each case was calculated using the above equation. The probability of a sequence occurring on a neuron was estimated by calculating the mean number of neurons in the postsynaptic population that received at least one sequence of that specific type somewhere along the dendrite.

## Estimating neuronal selectivity for the context of sequence computation

In order to estimate neuronal selectivity for ordered sequences, we stimulated the same set of ensembles in the presynaptic population in different temporal orderings, and measured the activation of neurons in the postsynaptic population. An input matrix was constructed, wherein the $M[i,j]^{th}$ entry was 1 if synapse j received an input at time step i. The local activation Q, at location x on the dendrite at time t, was modeled as:

$$Q\left(t,x\right) = \gamma Q\left(t-1,x\right) + \delta\left(t,x\right) + \psi\left(\sum_{x-\frac{S}{\sigma}-\frac{\Delta}{\sigma}}^{x-\frac{S}{\sigma}} \left(\left(1+\gamma Q\left(t-1,x\right)\delta\left(t,x\right)\right)^{\eta}-1\right)\right)$$

Here, the first term captures the decay of local activation in the absence of new input, i.e., γ<1. The second term represents the baseline contribution of inputs, (δ(t,x))=1 if an input arrived at location x at time t, else it is 0. The third term represents the nonlinear contribution of sequential input. Prior activation Q within a region of Δ a distance S away from location x is combined multiplicatively with input arriving at x at time t and is scaled nonlinearly using $\eta$. It is also passed $\phi$ which keeps the sequential contribution within bounds.

$$\psi\left(y\right) = 2V_{max}\left(\left(\frac{1}{1+e^{-cx}}\right)-0.5\right)$$

For the analysis in *Figure 7*, the parameters for this formulation were chosen such that in the absence of any ectopic inputs, the local selectivity for the perfectly ordered sequence was ~0.8. This value is close to the value we obtained for chemical selectivity shown in *Figure 5*.

The total activation of a neuron was obtained by integrating Q over all synapses across time to give the response $A_p$ for each pattern p. Selectivity for the ordered sequence is measured as

$$Selectivity = \frac{\left(A_{seq}-A_{mean}\right)}{A_{max}}$$

where $A_{seq}$ is the response to an ordered pattern, $A_{mean}$ was the mean of responses to all patterns, $A_{max}$ was the maximum response obtained among the set of patterns tested.

## Ca²⁺-Calmodulin simulation

With the help of MOOSE Simulator, we used the $Ca^{2+}$-CaM system to model selectivity for grouped inputs. The model involves $Ca^{2+}$ binding to Calmodulin in a series of steps resulting in activated $Ca^{2+}$-Calmodulin (*Figure 2a*). $Ca^{2+}$ stimulus was delivered in a spatially localized manner with synapses spaced at 2 μm, or in a dispersed manner, with inputs spaced at 10 μm on a 100 μm long dendrite

which had a diameter of 5 µm. The dendrite was discretized into one-dimensional voxels of length 0.5 µm. The mean concentration of CaM-Ca4 in the dendrite, wherein each CaM molecule is bound to four $Ca^{2+}$, was measured over time for the clustered and dispersed input cases. The parameters of the model have been provided in *Supplementary file 1a and b*.

## Chemical simulation - bistable-switch model

The bistable-switch model was adapted from *Bhalla, 2017*, which was available on ModelDB at https://modeldb.science/227318. We modified the original model to include an expression for parsing ectopic inputs. The original set of parameters from *Bhalla, 2017*, were retained for the bistable-switch model. Simulations were performed using MOOSE. For *Figure 5B*, inputs were delivered either as an ordered pattern [0, 1, 2, 3, 4] or a scrambled pattern [4, 1, 0, 3, 2] with consecutive inputs spaced at a distance of 3 µm on a 100 µm long dendrite that had a diameter of 10 µm. The time interval between inputs was 2 s. The directionality (Q score) used in *Figure 5C* was measured as $Q=mR^2$, where m is the slope and $R^2$ is the coefficient of determination of the linear regression fit of each pattern with respect to perfect ordering [0, 1, 2, 3, 4], as used in *Bhalla, 2017*. For *Figure 5E and F*, an additional set of 23 scrambled patterns obtained from different permutations of the perfectly ordered sequence were also tested. Since permutations of a sequence of length M can yield a total of M! patterns, we sub-sampled from this set to select every M!/(M–4)!th pattern to get a total of 24 patterns for M>3. Within this set the first pattern was the ordered sequence itself. For the case of M=3, only the six available patterns (=3! permutations) were used. Response was measured as the total amount of molecule A present in the dendrite summed over time. Using these responses selectivity was calculated with the help of *Equation 6*.

In *Figure 5E*, the reference selectivity was estimated using the above formula for a set of patterns made of five inputs each. Further, in addition to the inputs constituting a sequence, an ectopic input was delivered. Different spatiotemporal combinations of the position and time of arrival of the ectopic input were tested. For *Figure 5F*, we repeated this exercise for patterns of different lengths either without any ectopic (reference), with a single ectopic, or with two ectopic inputs. A smaller set of spatiotemporal pattern combinations was used in *Figure 5F* to keep the computational complexity within reasonable bounds. In addition, we restricted the analysis to ectopics occurring within the zone of the sequence.

## Effect of ectopic inputs on electrical sequences

We chose the model published on ModelDB (https://modeldb.science/140828) for examining the effect of ectopic inputs on the selectivity of electrical sequences (*Branco et al., 2010*). The base code from ModelDB was modified and run using the NEURON Simulator. The original channel kinetics parameters post the fix to NMDA_Mg_T initialization, and the dendrite used for testing were retained as is from the ModelDB GitHub repository (https://github.com/ModelDBRepository/140828, *Morse, 2020*). We removed the timing jitter between the inputs and positioned the inputs slightly closer (~11%) to avoid end-effects. We tested out the effects of ectopics using the passive version of the model.

For *Figure 6B*, a sequence of five inputs was delivered as an inward, outward, and scrambled sequence and the voltage response at the soma was measured. For *Figure 6C*, inputs were delivered sequentially and using a number of different scrambled orderings. The voltage response at the soma was measured in each case. Selectivity was calculated using *Equation 6*, wherein A represented the peak excitatory postsynaptic potentials in this case. This exercise was repeated for different time intervals between successive inputs and for different sequence lengths to obtain the selectivity vs input velocity curves.

For *Figure 6D and E* the base code was slightly modified to accommodate an ectopic input in addition to the different sequential/scrambled patterns that were being delivered. First, an ectopic input was added at different locations arriving at different times relative to the zone of the regular patterns, and selectivity is estimated at each position. The reference curve shows selectivity in the absence of the ectopic input.

This exercise was repeated for sequences of three to nine inputs with a subset of spatiotemporal combinations of the ectopic input. We sampled a combination of three locations (start, middle, and end) relative to the zone of the regular inputs, and three time points (start, middle, and end) for the

ectopic input. For sequences of ≥ 6 inputs, 121 pattern combinations of the regular inputs were sampled since sampling all permutations of the sequence was computationally expensive.

## Integrated multiscale model

We integrated each of the illustrated forms of spatial and temporal pattern selectivity into a single multiscale model that incorporated both electrical and chemical signaling. The model mechanisms and parameters were identical to those used in *Bhalla, 2017*, with a few small changes. First, this was a reduced geometry neuron with a Y-shaped dendritic structure, on which one arm of the Y was ornamented with 80 dendritic spines spaced at 2 µm. Second, the diffusion constant for CaM and its calcium-bound forms were set to 20 µm²/s. Third, the K_DR levels at the soma were up from 250 to 360 S/m². Fourth, the resting potential was lowered from –60 to –70 mV.

In brief, chemical signaling was present in the spines and dendrites, and these were coupled by diffusion. Chemical signaling and electrical signaling were also coupled, such that calcium influx through NMDA receptors and voltage-gated calcium channels led to signaling outcomes, and kinase activity led to channel phosphorylation which altered dendritic excitability. The model included six voltage-gated ion channels (Na, K_DR, K_A, K_Ca, K_AHP, L-type Ca) and three ligand-gated ion channels (AMPAR, NMDAR, and GABAR). AMPAR and NMDAR were located on the spines only, and GABAR was located on the dendritic branches at 16 locations on each branch. Each of the 32 GABA receptors received Poisson random synaptic input at a background rate of 1.0 Hz. The detailed list of parameters have been provided in *Supplementary file 1f and g*.

In order to run the simulations for a wide range of stimulus patterns, we implemented a wrapper script around the model named allNonlin7.py. This one script was used to perform all the simulations on the integrated model using the command line options specified in the *.bat files in the GitHub repository. Output from allNonlin7.py was saved as an hdf5 file and processed for visualization using readBackHdf7.py and analyzeNonlin6.py.

## Acknowledgements

BPS and USB are at NCBS-TIFR which receives the support of the Department of Atomic Energy, Government of India, under Project Identification No. RTI 4006. We would like to acknowledge NCBS Supercomputing Facility for access to the cluster. We would like to specially thank Mukund Thattai for ideas and discussions on the project. In addition, we acknowledge Anal Kumar, Sahil Moza, Shaurya Rahul Narlanka, Sriram Narayanan, Sulu Mohan, and Vinu Varghese Pulikkottil for valuable suggestions on the manuscript.

## Additional information

### Competing interests

Upinder Singh Bhalla: Reviewing editor, eLife. The other author declares that no competing interests exist.

### Funding

| Funder | Grant reference number | Author |
| --- | --- | --- |
| National Centre for Biological Sciences | | Bhanu Priya Somashekar |
| Department of Atomic Energy, Government of India | DAE Project Indentification Number- RTI4006 | Upinder Singh Bhalla |

The funders had no role in study design, data collection and interpretation, or the decision to submit the work for publication.

### Author contributions

Bhanu Priya Somashekar, Conceptualization, Formal analysis, Validation, Investigation, Visualization, Methodology, Writing – original draft, Writing – review and editing, Simulation, analysis and

visualization of - Figures 1,2,3,4,5,6,7; Upinder Singh Bhalla, Conceptualization, Formal analysis, Supervision, Funding acquisition, Investigation, Visualization, Methodology, Writing – original draft, Project administration, Writing – review and editing, Simulation and analysis - Figures 5, 8; and visualization of Figure 8

### Author ORCIDs
Bhanu Priya Somashekar ⓘ http://orcid.org/0000-0003-4873-767X
Upinder Singh Bhalla ⓘ https://orcid.org/0000-0003-1722-5188

Joint Public Review: https://doi.org/10.7554/eLife.100664.4.sa1
Author response https://doi.org/10.7554/eLife.100664.4.sa2

## Additional files

### Supplementary files
Supplementary file 1. Model parameters and equations. (a) Molecular species used in the $Ca^{2+}$-CaM model. (b) Reaction parameters of the $Ca^{2+}$-CaM model. (c) Kappa values for different kinds of groups. (d) p-Values for *Figure 5F*. (e) p-Values for *Figure 6E*. (f) Parameters of the multiscale model. (g) Channel distributions used in the multiscale model.

MDAR checklist

### Data availability
The current manuscript is a computational study, so all data can be generated from the code repository https://github.com/BhallaLab/synaptic-convergence-paper (copy archived at *Somashekar, 2025*). Instructions for generating the data for all figures are available on the readme file of the same respository.

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

# Appendix 1

Refer to *Table 2* in the main text for parameter definitions.

## Analytical derivations for different types of groups

### Probability of occurrence of stimulus-driven groups

Expectation number of connections from neurons in the M ensembles on a zone of length Z on the target neuron = $\frac{pNMZ}{L} = \nu_{cSDG}$.

Probability of a zone of length Z receiving M or more connections from any of the neurons in the M ensembles =

$$P_{cSDG\_Z} \approx 1 - \left( \sum_{m=0}^{m=M-1} \frac{\nu_{cSDG}^{m} e^{-\nu_{cSDG}}}{m!} \right)$$

Probability of a zone of length Z that receives at least M connections from any of the neurons in the M ensembles occurring anywhere along the dendritic length of a neuron,

$$P_{cSDG} \approx 1 - \left( 1 - P_{cSDG\_Z} \right)^{\kappa_{cSDG}}$$

where $\kappa_{cSDG}$ lies between $\frac{L}{Z}$ and $\frac{L}{\sigma}$ depending on the degree of overlap between zones in different network configurations.

Probability of a neuron receiving an *active stimulus-driven group* where there are at least M synapses receiving active inputs from neurons in the M ensembles is given by

$$P_{aSDG} \approx 1 - \left( 1 - P_{aSDG\_Z} \right)^{\kappa_{aSDG}}$$

where

$$P_{aSDG\_Z} \approx 1 - \left( \sum_{m=0}^{m=M-1} \frac{\nu_{SDG}^{a} e^{-\nu_{aSDG}}}{m!} \right) \text{ and } \nu_{aSDG} = \frac{pp_{e}NMZ}{L}$$

where $\kappa_{aSDG}$ again lies between $\frac{L}{Z}$ and $\frac{L}{\sigma}$ depending on the degree of overlap between zones.

### Probability of occurrence of noise groups

Consider a network where the rate of background activity at synapses is R Hz.

Expected number of inputs arriving in duration D due to background activity at a single synapse = $RD$.

Assuming that background activity in the presynaptic population is Poisson in nature, probability of a synapse being hit by background activity in duration D = $p_{bg} \approx 1 - e^{-RD}$.

Expected number of converging axonal connections in zone Z from the M ensembles = $\frac{pNMZ}{L}$.

Expected number of synapses being hit by noisy inputs in a dendritic zone of length Z in duration D = $\nu_{NG} \approx p_{bg} Z \left( \frac{1}{\sigma} - \frac{pNM}{L} \right)$

where $\sigma \approx$ inter-synapse interval.

Probability of inputs from background activity arriving at M or more synapses in a zone of length Z = $P_{NG\_Z} \approx 1 - \left( \sum_{m=0}^{m=M-1} \frac{\nu_{NG}^{m} e^{-\nu_{NG}}}{m!} \right)$.

Probability of a zone of length Z that has at least M synapses receiving background activity in duration D = $P_{NG} \approx 1 - \left( 1 - P_{NG\_Z} \right)^{\kappa_{NG}}$

where $\kappa_{NG}$ lies between $\frac{L}{Z}$ and $\frac{L}{\sigma}$ based on the degree of overlap between zones.

### Probability of occurrence of any group

Expected number of synapses being hit by noisy inputs in a dendritic zone of length Z in duration D = $\nu_{NG} \approx p_{bg} Z \left( \frac{1}{\sigma} - \frac{pNM}{L} \right)$.

Expected number of synapses in zone Z being hit by inputs from M ensembles = $\frac{pp_{e}NMZ}{L}$.

Expected number of synapses being hit either by ensemble inputs (from the M ensembles) or noisy inputs in a dendritic zone of length Z in duration D = $\nu_{AG} \approx p_{bg} Z \left( \frac{1}{\sigma} - \frac{pNM}{L} \right) + \frac{pp_{e}NMZ}{L}$.

Probability of either ensemble inputs or background activity or a combination of them arriving at M or more synapses in a zone of length Z = $P_{AG\_Z} \approx 1 - \left( \sum_{m=0}^{m=M-1} \frac{\nu_{AG}^m e^{-\nu_{AG}}}{m!} \right)$.

Probability of a zone of length Z that has at least M synapses receiving any activity in duration D = $P_{AG} \approx 1 - \left( 1 - P_{AG\_Z} \right)^{\kappa_{AG}}$

where $\kappa_{AG}$ lies between $\frac{L}{Z}$ and $\frac{L}{\sigma}$ depending on the degree of overlap between zones.

## Analytical derivations for different types of sequences

### Probability of occurrence of noise sequences

Assuming that background activity in the presynaptic population is Poisson in nature, probability of a synapse being hit by background activity in duration D = $p_{bg} \approx 1 - e^{-RD}$.

Expected number of synapses being hit by background activity from any neuron in the first time step D, anywhere on target neuron $\approx p_{bg} \left( \frac{L}{\sigma} - pN \right)$.

Expected number of inputs from background activity arriving in the second time step within a spacing of S to S+Δ from an input that arrived in the first time step = $\nu_{NS} \approx p_{bg} \left( \frac{L}{\sigma} - pN \right) \frac{\Delta}{L}$.

Expectation number of sequences of M inputs coming from background activity in successive time steps, with the subsequent input arriving within a spacing of S to S+Δ with respect to the preceding input = $E_{NS} \approx p_{bg} \left( \frac{L}{\sigma} - pN \right) \left( \nu_{NS} \right)^{M-1}$.

Using Poisson approximation, probability of one or more background activity-driven sequences of M inputs, occurring anywhere along the dendritic length of a neuron = $P_{NS} \approx 1 - e^{-E_{NS}}$.

### Probability of occurrence of any sequence

Expected number of synapses being hit by either background activity or from activity in neurons in the first ensemble in the first time step D, anywhere on target neuron $\approx pp_e N + p_{bg} \left( \frac{L}{\sigma} - pN \right)$.

Expected number of synapses being hit by either background activity or ensemble activity in the second time step within a spacing of S to S+Δ from an input that arrived in the first time step = $\nu_{AS} \approx \left( pp_e N + p_{bg} \left( \frac{L}{\sigma} - pN \right) \right) \frac{\Delta}{L}$.

Expected number of sequences of M inputs coming from either background activity or ensemble activity in successive time steps, with the subsequent input arriving within a spacing of S to S+Δ with respect to the preceding input = $E_{AS} \approx \left( pp_e N + p_{bg} \left( \frac{L}{\sigma} - pN \right) \right) \left( \nu_{AS} \right)^{M-1}$.

Using Poisson approximation, probability of one or more sequences of M inputs that are driven by either background inputs or ensemble inputs or a combination of them occurring anywhere along the dendritic length of a neuron = $P_{AS} \approx 1 - e^{-E_{AS}}$.

### Probability of occurrence of gap-fill sequences

Gap-fill sequences are those that are built from a combination of ensemble and background inputs. Hence, they contain at least one input from each of the two classes.

Expectation number of M-length gap-fill sequences constructed from inputs arriving in subsequent time windows of duration D, following the S to S+Δ spacing rule, on a neuron = $E_{GS} \approx E_{AS} - E_{aPOSS} - E_{NS}$.

Probability of occurrence of one or more gap-fill sequences containing M inputs anywhere on a neuron = $P_{GS} \approx 1 - e^{-E_{GS}}$.

