## [Editor Report · eLife Assessment]

This study presents **valuable** quantitative insights into the prevalence of functionally clustered synaptic inputs on neuronal dendrites. The simple analytical calculations and computer simulations provide **solid** support for the main arguments. The findings can lead to a more detailed understanding of how dendrites contribute to the computation of neuronal networks.

---

## [Referee Report · Joint Public Review]

Summary:

If synaptic input is functionally clustered on dendrites, nonlinear integration could increase the computational power of neural networks. But this requires the right synapses to be located in the right places. This paper aims to address the question of whether such synaptic arrangements could arise by chance (i.e. without special rules for axon guidance or structural plasticity), and could therefore be exploited even in randomly connected networks. This is important, particularly for the dendrites and biological computation communities, where there is a pressing need to integrate decades of work at the single-neuron level with contemporary ideas about network function.

Using an abstract model where ensembles of neurons project randomly to a postsynaptic population, back-of-envelope calculations are presented that predict the probability of finding clustered synapses and spatiotemporal sequences. Using data-constrained parameters, the authors conclude that clustering and sequences are indeed likely to occur by chance (for large enough ensembles), but require strong dendritic nonlinearities and low background noise to be useful.

Strengths:

- The back-of-envelope reasoning presented can provide fast and valuable intuition. The authors have also made the effort to connect the model parameters with measured values. Even an approximate understanding of cluster probability can direct theory and experiments towards promising directions, or away from lost causes.

- I found the general approach to be refreshingly transparent and objective. Assumptions are stated clearly about the model and statistics of different circuits. Along with some positive results, many of the computed cluster probabilities are vanishingly small, and noise is found to be quite detrimental in several cases. This is important to know, and I was happy to see the authors take a balanced look at conditions that help/hinder clustering, rather than just focus on a particular regime that works.

- This paper is also a timely reminder that synaptic clusters and sequences can exist on multiple spatial and temporal scales. The authors present results pertaining to the standard `electrical' regime (~50-100 µm, <50 ms), as well as two modes of chemical signaling (~10 µm, 100-1000 ms). The senior author is indeed an authority on the latter, and the simulations in Figure 5, extending those from Bhalla (2017), are unique in this area. In my view, the role of chemical signaling in neural computation is understudied theoretically, but research will be increasingly important as experimental technologies continue to develop.

(Editors' note: the paper has been through two rounds of revisions and the authors are encouraged to finalise this as the Version of Record. The earlier reviews are here: https://elifesciences.org/reviewed-preprints/100664v2/reviews)

---

## [Author Response]

The following is the authors’ response to the previous reviews.

**Reviewer #1 (Public Review):**
In this revision, the authors significantly improved the manuscript. They now address some of my concerns. Specifically, they show the contribution of end-effects on spreading the inputs between dendrites. This analysis reveals greater applicability of their findings to cortical cells, with long, unbranching dendrites than other neuronal types, such as Purkinje cells in the cerebellum.They now explain better the interactions between calcium and voltage signals, which I believe improve the take-away message of their manuscript. They modified and added new figures that helped to provide more information about their simulations.However, some of my points remain valid. Figure 6 shows depolarization of ~5mV from -75. This weak depolarization would not effectively recruit nonlinear activation of NMDARs. In their paper, Branco and Hausser (2010) showed depolarizations of ~10-15mV.More importantly, the signature of NMDAR activation is the prolonged plateau potential and activation at more depolarized resting membrane potentials (their Figure 4). Thus, despite including NMDARs in the simulation, the authors do not model functional recruitment of these channels. Their simulation is thus equivalent to AMPA only drive, which can indeed summate somewhat nonlinearly.

In the current study, we used short sequences of 5 inputs, since the convergence of longer sequences is extremely unlikely in the network configurations we have examined. This resulted in smaller EPSP amplitudes of ~5mV (Figure 6 - Supplement 2A, B). Longer sequences containing 9 inputs resulted in larger somatic depolarizations of ~10mV (Figure 6 - Supplement 2E, F). Although we had modified the (Branco, Clark, and Häusser 2010) model to remove the jitter in the timing of arrival of inputs and made slight modifications to the location of stimulus delivery on the dendrite, we saw similar amplitudes when we tested a 9-length sequence using (Branco, Clark, and Häusser 2010)’s published code (Figure 6 - Supplement 2I, J). In all the cases we tested (5 input sequence, 9 input sequence, 9 input sequence with (Branco, Clark, and Häusser 2010) code repository), removal of NMDA synapses lowered both the somatic EPSPs (Figure 6 - Supplement 2C,D,G,H,K,L) as well as the selectivity (measured as the difference between the EPSPs generated for inward and outward stimulus delivery) (Figure 6 Supplement 2M,N,O). Further, monitoring the voltage along the dendrite for a sequence of 5 inputs showed dendritic EPSPs in the range of 20-45 mV (Figure 6 - Supplement 2P, Q), which came down notably (10-25mV) when NMDA synapses were abolished (Figure 6 - Supplement 2R, S). Thus, even sequences containing as few as 5 inputs were capable of engaging the NMDA-mediated nonlinearity to show sequence selectivity, although the selectivity was not as strong as in the case of 9 inputs.

**Reviewer #1 (Recommendations for the authors):**
Minor points:Figure 8, what does the scale in A represent? I assume it is voltage, but there are no units. Figure 8, C, E, G, these are unconventional units for synaptic weights, usually, these are given in nS / per input.

We have corrected these. The scalebar in 8A represents membrane potential in mV. The units of 8C,E,G are now in nS.

**Reviewer #2 (Public Review):**
Summary:If synaptic input is functionally clustered on dendrites, nonlinear integration could increase the computational power of neural networks. But this requires the right synapses to be located in the right places. This paper aims to address the question of whether such synaptic arrangements could arise by chance (i.e. without special rules for axon guidance or structural plasticity), and could therefore be exploited even in randomly connected networks. This is important, particularly for the dendrites and biological computation communities, where there is a pressing need to integrate decades of work at the single-neuron level with contemporary ideas about network function.Using an abstract model where ensembles of neurons project randomly to a postsynaptic population, back-of-envelope calculations are presented that predict the probability of finding clustered synapses and spatiotemporal sequences. Using data-constrained parameters, the authors conclude that clustering and sequences are indeed likely to occur by chance (for large enough ensembles), but require strong dendritic nonlinearities and low background noise to be useful.Strengths:(1) The back-of-envelope reasoning presented can provide fast and valuable intuition. The authors have also made the effort to connect the model parameters with measured values. Even an approximate understanding of cluster probability can direct theory and experiments towards promising directions, or away from lost causes.(2) I found the general approach to be refreshingly transparent and objective. Assumptions are stated clearly about the model and statistics of different circuits. Along with some positive results, many of the computed cluster probabilities are vanishingly small, and noise is found to be quite detrimental in several cases. This is important to know, and I was happy to see the authors take a balanced look at conditions that help/hinder clustering, rather than to just focus on a particular regime that works.(3) This paper is also a timely reminder that synaptic clusters and sequences can exist on multiple spatial and temporal scales. The authors present results pertaining to the standard `electrical' regime (~50-100 µm, <50 ms), as well as two modes of chemical signaling (~10 µm, 100-1000 ms). The senior author is indeed an authority on the latter, and the simulations in Figure 5, extending those from Bhalla (2017), are unique in this area. In my view, the role of chemical signaling in neural computation is understudied theoretically, but research will be increasingly important as experimental technologies continue to develop.Weaknesses:(1) The paper is mostly let down by the presentation. In the current form, some patience is needed to grasp the main questions and results, and it is hard to keep track of the many abbreviations and definitions. A paper like this can be impactful, but the writing needs to be crisp, and the logic of the derivation accessible to non-experts. See, for instance, Stepanyants, Hof & Chklovskii (2002) for a relevant example.It would be good to see a restructure that communicates the main points clearly and concisely, perhaps leaving other observations to an optional appendix. For the interested but time-pressed reader, I recommend starting with the last paragraph of the introduction, working through the main derivation on page 7, and writing out the full expression with key parameters exposed. Next, look at Table 1 and Figure 2J to see where different circuits and mechanisms fit in this scheme. Beyond this, the sequence derivation on page 15 and biophysical simulations in Figures 5 and 6 are also highlights.

We appreciate the reviewers' suggestions. We have tightened the flow of the introduction. We understand that the abbreviations and definitions are challenging and have therefore provided intuitions and summaries of the equations discussed in the main text.

Clusters calculations

Our approach is to ask how likely it is that a given set of inputs lands on a short segment of dendrite, and then scale it up to all segments on the entire dendritic length of the cell.

Thus, the probability of occurrence of groups that receive connections from each of the M ensembles (PcFMG) is a function of the connection probability (p) between the two layers, the number of neurons in an ensemble (N), the relative zone-length with respect to the total dendritic arbor (Z/L) and the number of ensembles (M).

Sequence calculations

Here we estimate the likelihood of the first ensemble input arriving anywhere on the dendrite, and ask how likely it is that succeeding inputs of the sequence would arrive within a set spacing.

Thus, the probability of occurrence of sequences that receive sequential connections (PcPOSS) from each of the M ensembles is a function of the connection probability (p) between the two layers, the number of neurons in an ensemble (N), the relative window size with respect to the total dendritic arbor (Δ/L) and the number of ensembles (M).

(2) I wonder if the authors are being overly conservative at times. The result highlighted in the abstract is that 10/100000 postsynaptic neurons are expected to exhibit synaptic clustering. This seems like a very small number, especially if circuits are to rely on such a mechanism. However, this figure assumes the convergence of 3-5 distinct ensembles. Convergence of inputs from just 2 ense mbles would be much more prevalent, but still advantageous computationally. There has been excitement in the field about experiments showing the clustering of synapses encoding even a single feature.

We agree that short clusters of two inputs would be far more likely. We focused our analysis on clusters with three of more ensembles because of the following reasons:

(1) The signal to noise in these clusters was very poor as the likelihood of noise clusters is high.

(2) It is difficult to trigger nonlinearities with very few synaptic inputs.

(3) At the ensemble sizes we considered (100 for clusters, 1000 for sequences), clusters arising from just two ensembles would result in high probability of occurrence on all neurons in a network (~50% in cortex, see p_CMFG in figures below.). These dense neural representations make it difficult for downstream networks to decode (Foldiak 2003).

However, in the presence of ensembles containing fewer neurons or when the connection probability between the layers is low, short clusters can result in sparse representations (Figure 2 - Supplement 2). Arguments 1 and 2 hold for short sequences as well.

(3) The analysis supporting the claim that strong nonlinearities are needed for cluster/sequence detection is unconvincing. In the analysis, different synapse distributions on a single long dendrite are convolved with a sigmoid function and then the sum is taken to reflect the somatic response. In reality, dendritic nonlinearities influence the soma in a complex and dynamic manner. It may be that the abstract approach the authors use captures some of this, but it needs to be validated with simulations to be trusted (in line with previous work, e.g. Poirazi, Brannon & Mel, (2003)).

We agree that multiple factors might affect the influence of nonlinearities on the soma. The key goal of our study was to understand the role played by random connectivity in giving rise to clustered computation. Since simulating a wide range of connectivity and activity patterns in a detailed biophysical model was computationally expensive, we analyzed the exemplar detailed models for nonlinearity separately (Figures 5, 6, and new figure 8), and then used our abstract models as a proxy for understanding population dynamics. A complete analysis of the role played by morphology, channel kinetics and the effect of branching requires an in-depth study of its own, and some of these questions have already been tackled by (Poirazi, Brannon, and Mel 2003; Branco, Clark, and Häusser 2010; Bhalla 2017). However, in the revision, we have implemented a single model which incorporates the range of ion-channel, synaptic and biochemical signaling nonlinearities which we discuss in the paper (Figure 8, and Figure 8 Supplement 1, 2,3). We use this to demonstrate all three forms of sequence and grouped computation we use in the study, where the only difference is in the stimulus pattern and the separation of time-scales inherent in the stimuli.

(4) It is unclear whether some of the conclusions would hold in the presence of learning. In the signal-to-noise analysis, all synaptic strengths are assumed equal. But if synapses involved in salient clusters or sequences were potentiated, presumably detection would become easier? Similarly, if presynaptic tuning and/or timing were reorganized through learning, the conditions for synaptic arrangements to be useful could be relaxed. Answering these questions is beyond the scope of the study, but there is a caveat there nonetheless.

We agree with the reviewer. If synapses receiving connectivity from ensembles had stronger weights, this would make detection easier. Dendritic spikes arising from clustered inputs have been implicated in local cooperative plasticity (Golding, Staff, and Spruston 2002; Losonczy, Makara, and Magee 2008). Further, plasticity related proteins synthesized at a synapse undergoing L-LTP can diffuse to neighboring weakly co-active synapses, and thereby mediate cooperative plasticity (Harvey et al. 2008; Govindarajan, Kelleher, and Tonegawa 2006; Govindarajan et al. 2011). Thus if clusters of synapses were likely to be co-active, they could further engage these local plasticity mechanisms which could potentiate them while not potentiating synapses that are activated by background activity. This would depend on the activity correlation between synapses receiving ensemble inputs within a cluster vs those activated by background activity. We have mentioned some of these ideas in a published opinion paper (Pulikkottil, Somashekar, and Bhalla 2021). In the current study, we wanted to understand whether even in the absence of specialized connection rules, interesting computations could still emerge. Thus, we focused on asking whether clustered or sequential convergence could arise even in a purely randomly connected network, with the most basic set of assumptions. We agree that an analysis of how selectivity evolves with learning would be an interesting topic for further work.

**References**

Bhalla, Upinder S. 2017. “Synaptic Input Sequence Discrimination on Behavioral Timescales Mediated by Reaction-Diffusion Chemistry in Dendrites.” Edited by Frances K Skinner. *eLife* 6 (April):e25827. https://doi.org/10.7554/eLife.25827.Branco, Tiago, Beverley A. Clark, and Michael Häusser. 2010. “Dendritic Discrimination of Temporal Input Sequences in Cortical Neurons.” *Science (New York, N.Y.)* 329 (5999): 1671–75. https://doi.org/10.1126/science.1189664.Foldiak, Peter. 2003. “Sparse Coding in the Primate Cortex.” *The Handbook of Brain Theory and Neural Networks*. https://research-repository.st-andrews.ac.uk/bitstream/handle/10023/2994/FoldiakSparse HBTNN2e02.pdf?sequence=1.Golding, Nace L., Nathan P. Staff, and Nelson Spruston. 2002. “Dendritic Spikes as a Mechanism for Cooperative Long-Term Potentiation.” *Nature* 418 (6895): 326–31. https://doi.org/10.1038/nature00854.Govindarajan, Arvind, Inbal Israely, Shu-Ying Huang, and Susumu Tonegawa. 2011. “The Dendritic Branch Is the Preferred Integrative Unit for Protein Synthesis-Dependent LTP.” *Neuron* 69 (1): 132–46. https://doi.org/10.1016/j.neuron.2010.12.008.Govindarajan, Arvind, Raymond J. Kelleher, and Susumu Tonegawa. 2006. “A Clustered Plasticity Model of Long-Term Memory Engrams.” *Nature Reviews Neuroscience* 7 (7): 575–83. https://doi.org/10.1038/nrn1937.Harvey, Christopher D., Ryohei Yasuda, Haining Zhong, and Karel Svoboda. 2008. “The Spread of Ras Activity Triggered by Activation of a Single Dendritic Spine.” *Science (New York, N.Y.)* 321 (5885): 136–40. https://doi.org/10.1126/science.1159675.Losonczy, Attila, Judit K. Makara, and Jeffrey C. Magee. 2008. “Compartmentalized Dendritic Plasticity and Input Feature Storage in Neurons.” *Nature* 452 (7186): 436–41. https://doi.org/10.1038/nature06725.Poirazi, Panayiota, Terrence Brannon, and Bartlett W. Mel. 2003. “Pyramidal Neuron as Two-Layer Neural Network.” *Neuron* 37 (6): 989–99. https://doi.org/10.1016/S0896-6273(03)00149-1.Pulikkottil, Vinu Varghese, Bhanu Priya Somashekar, and Upinder S. Bhalla. 2021. “Computation, Wiring, and Plasticity in Synaptic Clusters.” *Current Opinion in Neurobiology*, Computational Neuroscience, 70 (October):101–12. https://doi.org/10.1016/j.conb.2021.08.001.